# Optimizing Language Models for Crosslingual Knowledge Consistency

## Abstract

Large language models are known to often exhibit inconsistent knowledge. This is particularly problematic in multilingual scenarios, where models are likely to be asked similar questions in different languages, and inconsistent responses can undermine their reliability. In this work, we show that this issue can be mitigated using reinforcement learning with a structured reward function, which leads to an optimal policy with consistent crosslingual responses. We introduce Direct Consistency Optimization (DCO), a DPO-inspired method that requires no explicit reward model and is derived directly from the LLM itself. Comprehensive experiments show that DCO significantly improves crosslingual consistency across diverse LLMs and outperforms existing methods when training with samples of multiple languages, while complementing DPO when gold labels are available. Extra experiments demonstrate the effectiveness of DCO in bilingual settings, significant out-of-domain generalizability, and controllable alignment via direction hyperparameters. Taken together, these results establish DCO as a robust and efficient solution for improving knowledge consistency across languages in multilingual LLMs. All code, training scripts, and evaluation benchmarks are released at `https://anonymous`.

## 1 Introduction

As multilingual capabilities become a standard feature of modern large language models (LLMs) (Touvron et al., 2023; OpenAI et al., 2023; DeepSeek-AI et al., 2025), ensuring crosslingual consistency (CLC) has become increasingly critical. Ideally, an LLM should provide consistent answers to the same factual question regardless of the language in which it is asked. However, this is far from guaranteed: prior work (Jiang et al., 2020; Qi et al., 2023; Wang et al., 2025b) has revealed that LLMs often produce conflicting responses across languages, as shown in Fig. 1 (left). Such inconsistencies can confuse users with diverse language backgrounds and undermine trust in multilingual systems.

To address this challenge, we aim to improve CLC by Reinforcement Learning (RL), inspired by principles of alignment with human preferences. Existing post-training algorithms for aligning with human preferences, such as proximal policy optimization (PPO; Schulman et al., 2017) and direct preference optimization (DPO; Ouyang et al., 2022; Rafailov et al., 2023), rely on reward functions defined over pairs of responses, often modeled using the Bradley–Terry framework. However, CLC involves connecting multiple languages and requires a different approach to reward design and optimization.

To this end, we propose a new reward function that promotes CLC by leveraging the likelihoods assigned by a model to the same answer expressed in different languages. Specifically, to align two languages, $L_1$ and $L_2$, we define the reward for $L_1$ based on the log-likelihood of answers generated when prompted in $L_2$, and vice versa. This leads to a policy expressed as a *product of experts* (Hinton, 1999): $\pi^\star(\boldsymbol{y}^i \mid \boldsymbol{x}^i) = \frac{1}{Z} \prod_j \left( \pi_{\text{REF}}\big(\tau^j(\boldsymbol{y}^i) \mid \tau^j(\boldsymbol{x}^i)\big) \right)^{w_{ij}}$, where $\pi_{\text{REF}}$ is the base multilingual LLM, $\tau^i$ translates a prompt or response to $L_i$, $w_{ij}$ are controllable parameters, and $Z$ is a normalization term. By controlling $w_{ij}$, the optimal policy $\pi^\star$ can be theoretically guaranteed to be consistent while preserving the model's overall performance. Building on this foundation, we propose **direct consistency optimization** (DCO), an efficient algorithm that adapts a DPO-like procedure to our proposed objective without explicit reward models. We theoretically prove that DCO can bypass the online sampling step in RL, yet still arrive at the same optimal policy as the original RL formulation.

Figure 1: Illustration of DCO, which promotes crosslingual consistency by aligning the likelihood of completions across parallel prompts in different languages.

As illustrated in Fig. 1, training involves parallel prompts and completions across languages (middle), and the optimization encourages consistent distributions over completions in both languages (right).

We evaluate the effectiveness of the proposed reward and optimization algorithm with nine LLMs across three datasets, covering 26 languages. Experimental results demonstrate that DCO significantly improves crosslingual consistency while maintaining, and often improving, answer accuracy in the post-trained languages.

In summary, our contributions are as follows:

- We propose a reward function tailored for CLC and introduce DCO, an algorithm that solves the RL objective, with theoretical guarantees of improved CLC and preserved task performance.
- We empirically validate DCO on advanced LLMs across diverse benchmarks.
- We provide extensive analyses, including comparisons with other alignment techniques, cross-domain generalization, and control over language preference.

## 2 RELATED WORK

Crosslingual knowledge consistency is a crucial property for multilingual LLMs.

**Measuring CLC.** Several studies have explored methods to assess the consistency of knowledge in multilingual LLMs. Xing et al. (2024) and Ai et al. (2025) evaluate consistency by measuring the agreement of top-1 generated answers to the same question posed in different languages, whereas Jiang et al. (2020) compute the average overlapping ratio of correct predictions across languages. To assess CLC more comprehensively and to disentangle it from accuracy, Qi et al. (2023) introduce the RankC metric, based on a weighted average of the overlapping ratio of top-1 to top-N ranked candidates. These studies reveal significant crosslingual knowledge inconsistencies in a wide range of LLMs.

**Improving CLC.** A number of recent studies attempt to improve CLC by applying vector interventions on the hidden representations of LLMs (Lu et al., 2025; Wang et al., 2025a; Liu & Niehues, 2025). While promising and insightful, these interpretability-based methods are typically tested on small datasets and specific models, making them challenging to scale to broader applications.

Closer to our work, Wang et al. (2025b) proposes CALM, which improves CLC using RL. Their approach first selects a target answer by majority voting based on the model's completions across multiple languages. Then, they use DPO to increase the likelihood of the target majority answer across languages. However, CALM requires more than two languages, restricting its usage in practical bilingual scenarios. Moreover, it does not necessarily benefit from adding more languages as majority voting can become unreliable when multiple low-resource languages are included.

## 3 PRELIMINARIES

**Reinforcement Learning from Human Feedback.** Reinforcement Learning from Human Feedback (RLHF) typically starts with **supervised fine-tuning** (SFT) a pre-trained language model $\pi_{\boldsymbol{\theta}}$ on a dataset $\mathcal{D}_{\text{SFT}}$ containing annotated examples for downstream tasks to minimize the loss $\mathcal{L}^{\text{ML}}(\boldsymbol{\theta}) = -\mathbb{E}_{(\boldsymbol{x}, \boldsymbol{y}) \sim \mathcal{D}_{\text{SFT}}} \big[ \log \pi_{\boldsymbol{\theta}}(\boldsymbol{y} \mid \boldsymbol{x}) \big]$. The resulting fine-tuned model is denoted by $\pi_{\text{SFT}}$. However, it may not always reflect human preferences. This misalignment arises because the maximum likelihood estimation objective does not differentiate between major and minor errors in the model's responses. To address this, a **reward optimization** step is introduced. Assuming the availability of a reward model $r(\boldsymbol{x}, \boldsymbol{y})$, which is trained on a dataset with human feedback $\mathcal{D}_{\text{HF}}$, the RLHF objective aims to maximize the expected reward of the model's outputs. A KL divergence regularization term is added to the objective (Stiennon et al., 2020) to prevent reward hacking (Amodei et al., 2016) and ensure the model does not deviate excessively from $\pi_{\text{SFT}}$. The ultimate target is to obtain a $\pi_{\boldsymbol{\theta}}$:

$$\max_{\pi_{\boldsymbol{\theta}}} \mathbb{E}_{\boldsymbol{x} \sim \mathcal{D}, \boldsymbol{y} \sim \pi_{\boldsymbol{\theta}}(\cdot | \boldsymbol{x})} \Big[ r(\boldsymbol{x}, \boldsymbol{y}) \Big] - \beta \cdot \text{KL} \big[ \pi_{\boldsymbol{\theta}}(\boldsymbol{y} \mid \boldsymbol{x}) \, \| \, \pi_{\text{SFT}}(\boldsymbol{y} \mid \boldsymbol{x}) \big], \tag{1}$$

where $\beta$ is a hyperparameter controlling the adherence to $\pi_{\text{SFT}}$. This objective is typically optimized using algorithms such as PPO or other actor–critic methods (Mnih et al., 2016; Glaese et al., 2022). As proposed by Rafailov et al. 2023, the optimal $\pi^{\star}$ can be expressed in the closed form:

$$\pi^{\star}(\boldsymbol{y} \mid \boldsymbol{x}) = \frac{1}{Z(\boldsymbol{x})} \pi_{\text{SFT}}(\boldsymbol{y} \mid \boldsymbol{x}) \exp\left( \frac{1}{\beta} r(\boldsymbol{x}, \boldsymbol{y}) \right). \tag{2}$$

where $Z(\boldsymbol{x})$ is the partition function that ensures normalization.

**Direct Preference Optimization.** Rafailov et al. (2023) observe that by rearranging the terms in Eq. (2), the reward function $r$ can be reparameterized as:

$$r(\boldsymbol{x}, \boldsymbol{y}) = \beta \log \frac{\pi^{\star}(\boldsymbol{y} \mid \boldsymbol{x})}{\pi_{\text{REF}}(\boldsymbol{y} \mid \boldsymbol{x})} + \beta \log Z(\boldsymbol{x}), \tag{3}$$

where $\pi_{\text{REF}}$ is the reference policy. To avoid training reward models, Rafailov et al. (2023) propose DPO, which directly optimizes the policy $\pi_{\boldsymbol{\theta}}$ with the following loss function:

$$\mathcal{L}^{\text{DPO}}(\boldsymbol{\theta}) = - \mathbb{E}_{(\boldsymbol{x}, \boldsymbol{y}_w, \boldsymbol{y}_l) \sim \mathcal{D}_{\text{HF}}} \Big[ \log \mathbb{P}_{\widehat{r}_{\boldsymbol{\theta}}}(\boldsymbol{y}_w \succ \boldsymbol{y}_l) \Big], \tag{4}$$

where an estimated reward $\widehat{r}_{\boldsymbol{\theta}} \overset{\text{def}}{=} \beta \log \frac{\pi_{\boldsymbol{\theta}}(\boldsymbol{y}|\boldsymbol{x})}{\pi_{\text{REF}}(\boldsymbol{y}|\boldsymbol{x})}$ replaces the true reward $r$ and $(\boldsymbol{y}_w, \boldsymbol{y}_l)$ represents a pair of preferred and dispreferred responses (or 'winning' and 'loosing', respectively). Minimizing Eq. (4) yields the *same* optimal policy $\pi^{\star}$ as optimizing Eq. (1) with a reward function trained on $\mathcal{D}_{\text{HF}}$ (Rafailov et al., 2023, Theorem 1; Azar et al., 2023, Proposition 4).

## 4 OPTIMIZING CROSSLINGUAL CONSISTENCY

In this section, we formulate the problem of alignment for consistency as an RL task, define the reward function, and derive the optimal policy that ensures consistent responses across languages.

### 4.1 DEFINING CROSSLINGUAL CONSISTENCY

Throughout this paper, we use superscripts to denote the language of a prompt $\boldsymbol{x}$ or a response $\boldsymbol{y}$. For example, $\boldsymbol{x}^1$ represents a prompt in language $L_1$, $\boldsymbol{y}^2$ represents a response in language $L_2$. We define two prompt–response pairs $(\boldsymbol{x}^1, \boldsymbol{y}^1)$ and $(\boldsymbol{x}^2, \boldsymbol{y}^2)$ in languages $L_1$ and $L_2$ as **equivalent**, denoted by $(\boldsymbol{x}^1, \boldsymbol{y}^1) \sim (\boldsymbol{x}^2, \boldsymbol{y}^2)$, if they can be mapped to each other via translational mappings $\tau^1 \colon L_2 \to L_1$ and $\tau^2 \colon L_1 \to L_2$.[1] For simplicity, we denote $\tau^1$, which maps strings from $L_2$ to $L_1$, to be the inverse of $\tau^2$. For example, this implies $\boldsymbol{x}^2 = \tau^2(\boldsymbol{x}^1)$ and $\boldsymbol{y}^1 = \tau^1(\boldsymbol{y}^2)$.

---

[1]We assume the existence of such translational mappings $\tau^1, \tau^2$, particularly in factual question-answering settings where the answers to a question are objective and the set of candidate answers is finite. Prior work on zero-shot crosslingual transfer and reward transfer (Wu & Dredze, 2019; Wu et al., 2024) discussed the generalizability of this assumption to other fields that involve open-ended generation.

We formalize **crosslingual consistency** as the property that the relative preference between any pair of responses remains *unchanged across different languages*.

**Definition 1.** *A language model $\pi^\star$ is **consistent** across $L_1$ and $L_2$ if*

$$\pi^\star(\boldsymbol{y}_w^1 \mid \boldsymbol{x}^1) \geq \pi^\star(\boldsymbol{y}_l^1 \mid \boldsymbol{x}^1) \iff \pi^\star(\boldsymbol{y}_w^2 \mid \boldsymbol{x}^2) \geq \pi^\star(\boldsymbol{y}_l^2 \mid \boldsymbol{x}^2). \tag{5}$$

*for all $(\boldsymbol{x}^1, \boldsymbol{y}_w^1) \sim (\boldsymbol{x}^2, \boldsymbol{y}_w^2)$ and $(\boldsymbol{x}^1, \boldsymbol{y}_l^1) \sim (\boldsymbol{x}^2, \boldsymbol{y}_l^2)$.*

In other words, given a prompt $\boldsymbol{x}$ and a pair of responses $(\boldsymbol{y}_w, \boldsymbol{y}_l)$, a consistent language LLM should maintain the same preference for one response over the other, regardless of the language in which the prompt and responses are expressed. See App. C for the rationale behind not enforcing exact distribution matching in Def. 1.

## 4.2 A PIECEWISE REWARD FUNCTION FOR CONSISTENCY

To align a model $\pi_{\text{REF}}$ across $L_1$ and $L_2$, we propose the following **piecewise reward function**:

$$r_{\text{ALIGN}}(\boldsymbol{x}, \boldsymbol{y}) = \begin{cases} \gamma_1 \log \pi_{\text{REF}}(\tau^2(\boldsymbol{y}) \mid \tau^2(\boldsymbol{x})) & \text{if } \boldsymbol{x}, \boldsymbol{y} \in L_1, \\ \gamma_2 \log \pi_{\text{REF}}(\tau^1(\boldsymbol{y}) \mid \tau^1(\boldsymbol{x})) & \text{if } \boldsymbol{x}, \boldsymbol{y} \in L_2, \\ 0 & \text{otherwise}, \end{cases} \tag{6}$$

where $\gamma_1, \gamma_2 \in \mathbb{R}^+$ are parameters controlling the deviation from $\pi_{\text{REF}}$ in each language. A smaller $\gamma_1$ keeps the aligned model closer to $\pi_{\text{REF}}$ in $L_1$, while a smaller $\gamma_2$ does the same for $L_2$. Note that $\gamma_1$ and $\gamma_2$ are distinct from $\beta$ (see Eq. (1)), which controls the *overall* deviation of the target policy from the base policy $\pi_{\text{REF}}$.

## 4.3 SOLVING THE CONSTRAINED RL PROBLEM

Substituting $r_{\text{ALIGN}}$ back into the RL objective Eq. (1), the RL problem becomes:

$$\max_{\pi_{\boldsymbol{\theta}}} \mathbb{E}_{\boldsymbol{x} \sim \mathcal{D}, \boldsymbol{y} \sim \pi_{\boldsymbol{\theta}}(\cdot \mid \boldsymbol{x})} \left[ r_{\text{ALIGN}}(\boldsymbol{x}, \boldsymbol{y})) \right] - \beta \cdot \text{KL} \left[ \pi_{\boldsymbol{\theta}}(\cdot \mid \boldsymbol{x}) \| \pi_{\text{REF}}(\cdot \mid \boldsymbol{x}) \right], \tag{7}$$

where $\mathcal{D}$ is a set of factual question prompts. The optimal policy is given by Rafailov et al., 2023:

$$\pi^\star(\boldsymbol{y} \mid \boldsymbol{x}) = \frac{1}{Z(\boldsymbol{x})} \pi_{\text{REF}}(\boldsymbol{y} \mid \boldsymbol{x}) \exp \left( \frac{1}{\beta} r_{\text{ALIGN}}(\boldsymbol{x}, \boldsymbol{y}) \right), \tag{8}$$

where $Z(\boldsymbol{x})$ is the normalization constant. For $L_1$ and $L_2$, the resulting policy takes a product-of-experts form (Hinton, 1999), which expands to:

$$\pi^\star(\boldsymbol{y}^1 \mid \boldsymbol{x}^1) = \frac{1}{Z(\boldsymbol{x}^1)} \pi_{\text{REF}}(\boldsymbol{y}^1 \mid \boldsymbol{x}^1) \left( \pi_{\text{REF}}(\tau^2(\boldsymbol{y}^1) \mid \tau^2(\boldsymbol{x}^1)) \right)^{\gamma_1/\beta}, \quad (\boldsymbol{x}^1, \boldsymbol{y}^1 \in L_1) \tag{9a}$$

$$\pi^\star(\boldsymbol{y}^2 \mid \boldsymbol{x}^2) = \frac{1}{Z(\boldsymbol{x}^2)} \pi_{\text{REF}}(\boldsymbol{y}^2 \mid \boldsymbol{x}^2) \left( \pi_{\text{REF}}(\tau^1(\boldsymbol{y}^2) \mid \tau^1(\boldsymbol{x}^2)) \right)^{\gamma_2/\beta}. \quad (\boldsymbol{x}^2, \boldsymbol{y}^2 \in L_2) \tag{9b}$$

**Choosing $\gamma_1, \gamma_2$ and $\beta$.** *Not all* combinations of $\gamma_1$, $\gamma_2$, and $\beta$ guarantee a consistent optimal policy. Lemma 1 provides a condition for selecting these hyperparameters to ensure consistency.

**Lemma 1.** *If $\gamma_1 \gamma_2 = \beta^2$, the optimal policy $\pi^\star$ defined by Eq. (8) is consistent across $L_1$ and $L_2$.*

*Proof sketch.* When $\gamma_1 \gamma_2 = \beta^2$, raising both sides of Eq. (9a) to the power of $\frac{\beta}{\gamma_1}$ gives us:

$$\left( \pi^\star(\boldsymbol{y}^1 \mid \boldsymbol{x}^1) \right)^{\beta/\gamma_1} \equiv \frac{Z(\tau^2(\boldsymbol{x}^1))}{Z^{\beta/\gamma_1}(\boldsymbol{x}^1)} \pi^\star(\tau^2(\boldsymbol{y}^1) \mid \tau^2(\boldsymbol{x}^1)).$$

Since the function $f(x) = cx^{\beta/\gamma_1}$ increases monotonically in $x$ for $\beta/\gamma_1 > 0, c > 0$, we have $\pi^\star(\boldsymbol{y}_w^1 \mid \boldsymbol{x}^1) \geq \pi^\star(\boldsymbol{y}_l^1 \mid \boldsymbol{x}^1) \iff \pi^\star(\tau^2(\boldsymbol{y}_w^1) \mid \tau^2(\boldsymbol{x}^1)) \geq \pi^\star(\tau^2(\boldsymbol{y}_l^1) \mid \tau^2(\boldsymbol{x}^1))$, for all $\boldsymbol{y}_w^1, \boldsymbol{y}_l^1$. Thus, $\pi^\star$ is consistent across $L_1$ and $L_2$. See App. D.1 for details. ∎

**Remark 1.** *Optimizing the objective in Eq. (7) yields a policy $\pi^\star$ that balances the original policy $\pi_{\text{REF}}$ across $L_1$ and $L_2$. Lemma 1 specifies the relationship $\gamma_1 \gamma_2 = \beta^2$ to ensure consistency. Here, $\beta$ controls the overall deviation of $\pi^\star$ from $\pi_{\text{REF}}$. While $\gamma_1$ and $\gamma_2$ determine the relative alignment strength for $L_1$ and $L_2$. For instance, a smaller $\gamma_1$ biases $\pi^\star$ closer to $\pi_{\text{REF}}$ in $L_1$.*

**What does $r_{\text{ALIGN}}$ do?** Maximizing $\mathbb{E}_{\boldsymbol{y} \sim \pi_{\boldsymbol{\theta}}(\cdot|\boldsymbol{x})}\left[r_{\text{ALIGN}}(\boldsymbol{x}, \boldsymbol{y})\right]$ is equivalent to maximizing the weighted summation $\sum_{\boldsymbol{y}} \pi_{\boldsymbol{\theta}}(\boldsymbol{y} \mid \boldsymbol{x}) r_{\text{ALIGN}}(\boldsymbol{x}, \boldsymbol{y})$. According to the **rearrangement inequality**, this summation achieves its maximum when the sequences $\{\pi_{\boldsymbol{\theta}}(\boldsymbol{y} \mid \boldsymbol{x})\}_{\boldsymbol{y}}$ and $\{r_{\text{ALIGN}}(\boldsymbol{x}, \boldsymbol{y})\}_{\boldsymbol{y}}$ are *monotonically aligned* (Hardy et al., 1952). This alignment directly corresponds to our notion of **consistency**. By defining $r_{\text{ALIGN}}$ as in Eq. (6), which reflects the likelihood of a response in *the other language*, $\pi_{\boldsymbol{\theta}}$ is encouraged to align its preferences across languages, thereby promoting consistency.

**Generalizing to $N$ languages.** Our method naturally extends to align $N$ languages. For $N$ languages, we introduce $(N^2 - N)$ hyperparameters $\gamma_{ij}$, where $i, j \in \{1, 2, \ldots, N\}$ and $i \neq j$, to control the alignment strength between $L_i$ and $L_j$. The reward function is $r_{\text{ALIGN}}(\boldsymbol{x}, \boldsymbol{y}) = \sum_{j=1}^{N} \gamma_{ij} \log \pi_{\text{REF}}(\tau^j(\boldsymbol{y}) \mid \tau^j(\boldsymbol{x}))$ when $\boldsymbol{x}, \boldsymbol{y} \in L_i$. and the optimal policy is given by:

$$\pi^\star(\boldsymbol{y}^i \mid \boldsymbol{x}^i) = \frac{1}{Z(\boldsymbol{x}^i)} \pi_{\text{REF}}(\boldsymbol{y}^i \mid \boldsymbol{x}^i) \prod_{j \neq i \wedge j \in \{1, 2, \cdots, N\}} \left(\pi_{\text{REF}}\left(\tau^j(\boldsymbol{y}^i) \mid \tau^j(\boldsymbol{x}^i)\right)\right)^{\gamma_{ij}/\beta}, \quad (10)$$

where $\boldsymbol{x}^i, \boldsymbol{y}^i \in L_i$ and $Z(\boldsymbol{x}^i)$ is the normalization constant, for $i \in \{1, 2, \ldots, N\}$.

The detailed derivation and the constraints on $\gamma_{ij}$ to ensure consistency are provided in App. E. This formulation ensures that the policy $\pi^\star$ aligns preferences across all $N$ languages while maintaining flexibility through the hyperparameters $\gamma_{ij}$.

## 4.4 DIRECT CONSISTENCY OPTIMIZATION

In principle, there could be diverse ways to implement $r_{\text{ALIGN}}$. Here, we propose Direct Consistency Optimization (DCO) as an efficient algorithm tailored to our consistency objective. DCO is inspired by DPO, which bypasses the reward modeling and constrained RL phase. It leverages a dataset of parallel prompt–response pairs, eliminating the need for online sampling and translator usage.

**The Objective Function.** The core idea of DPO is to use a change of variables to express the human preference alignment loss directly as a function of the policy $\pi_{\boldsymbol{\theta}}$. In Eq. (6), we have described the exact form of $r_{\text{ALIGN}}$ that we need. Our goal is to design an objective function that will lead to an optimal $\widehat{r}_{\boldsymbol{\theta}}$ that is the same as $r_{\text{ALIGN}}$, and thus leads to policy $\pi_{\boldsymbol{\theta}}$ that is the same as $\pi^\star$. In principle, any objective function with an optimal solution of $r_{\text{ALIGN}}$ can be used. Here, we adopt an objective that mirrors the DPO framework, leveraging the Bradley–Terry preference model to align reward differences with expected values. Specifically, we train $\widehat{r}_{\boldsymbol{\theta}}(\boldsymbol{x}, \boldsymbol{y}_w) - \widehat{r}_{\boldsymbol{\theta}}(\boldsymbol{x}, \boldsymbol{y}_l)$ to match $r_{\text{ALIGN}}(\boldsymbol{x}, \boldsymbol{y}_w) - r_{\text{ALIGN}}(\boldsymbol{x}, \boldsymbol{y}_l)$. Through this modeling choice, we avoid computing the intractable normalization term $Z(\boldsymbol{x})$ in Eq. (3).

Let $\mathcal{D}_\parallel$ denote a dataset of parallel prompt–response pairs, from which we sample tuples $(\boldsymbol{x}^1, \boldsymbol{y}^1, \boldsymbol{x}^2, \boldsymbol{y}^2)$, where $(\boldsymbol{x}^1, \boldsymbol{y}^1) \sim (\boldsymbol{x}^2, \boldsymbol{y}^2)$. The responses are *randomly* paired into $\boldsymbol{y}_w^1, \boldsymbol{y}_l^1$, meaning we do not assume $\boldsymbol{y}_w^1$ is inherently better than $\boldsymbol{y}_l^1$. We define the following loss function to train $\widehat{r}_{\boldsymbol{\theta}}$ to match $r_{\text{ALIGN}}$:

$$L(\boldsymbol{\theta}) = \mathbb{E}_{(\boldsymbol{x}^1, \boldsymbol{y}_w^1, \boldsymbol{y}_l^1, \boldsymbol{x}^2, \boldsymbol{y}_w^2, \boldsymbol{y}_l^2) \sim \mathcal{D}_\parallel} \left[\left\|\left(\widehat{r}_{\boldsymbol{\theta}}(\boldsymbol{x}^1, \boldsymbol{y}_w^1) - \widehat{r}_{\boldsymbol{\theta}}(\boldsymbol{x}^1, \boldsymbol{y}_l^1)\right) - \gamma_1 \log \frac{\pi_{\text{REF}}(\boldsymbol{y}_w^2 \mid \boldsymbol{x}^2)}{\pi_{\text{REF}}(\boldsymbol{y}_l^2 \mid \boldsymbol{x}^2)}\right\| + \right.$$
$$\left. \left\|\left(\widehat{r}_{\boldsymbol{\theta}}(\boldsymbol{x}^2, \boldsymbol{y}_w^2) - \widehat{r}_{\boldsymbol{\theta}}(\boldsymbol{x}^2, \boldsymbol{y}_l^2)\right) - \gamma_2 \log \frac{\pi_{\text{REF}}(\boldsymbol{y}_w^1 \mid \boldsymbol{x}^1)}{\pi_{\text{REF}}(\boldsymbol{y}_l^1 \mid \boldsymbol{x}^1)}\right\|\right]. \quad (11)$$

Minimizing Eq. (11) yields the same optimal policy as Eq. (7), as formalized in the following lemma:

**Lemma 2.** *When Eq. (11) is minimized, the reward function $\widehat{r}_{\boldsymbol{\theta}}$ will converge to*

$$\widehat{r}_{\boldsymbol{\theta}}^\star(\boldsymbol{x}, \boldsymbol{y}) = \begin{cases} \gamma_1 \log \pi_{\text{REF}}(\tau^2(\boldsymbol{y}) \mid \tau^2(\boldsymbol{x})) + c(\boldsymbol{x}) & \text{if } \boldsymbol{x}, \boldsymbol{y} \in L_1, \\ \gamma_2 \log \pi_{\text{REF}}(\tau^1(\boldsymbol{y}) \mid \tau^1(\boldsymbol{x})) + c(\boldsymbol{x}) & \text{if } \boldsymbol{x}, \boldsymbol{y} \in L_2, \end{cases} \quad (12)$$

*where $c(\boldsymbol{x})$ is a function independent of $\boldsymbol{y}$.*

See App. D.2 for proof. By combining Lemma 2 with Rafailov et al. (2023, Theorem 1), we conclude that a consistent policy can be directly trained without explicitly training a reward function $r$ or solving a constrained RL problem. We further compare our method with DPO in App. F.

## 5 EXPERIMENTAL SETUP

**Models.** We evaluate our method on 9 multilingual models from 4 LLM families with sizes ranging from 4B to 14B, namely: `Qwen2.5-7B/14B` (Qwen et al., 2025), `Qwen3-8B/14B` (Yang et al., 2025), `Aya-Expanse-8B` (Üstün et al., 2024), `Llama3.1-8B`, `Llama3.2-3B` (Dubey et al., 2024), and `Gemma3-4B/12B` (Kamath et al., 2025). Training configurations are provided in App. G.

**Datasets.** We use three different multilingual question answering benchmarks: MMMLU (Hendrycks et al., 2021), XCSQA (Lin et al., 2021), and BMLAMA (Qi et al., 2023). All three contain parallel questions and candidate completions over all tested languages, translated from their English origin. MMMLU is a multilingual extension of the MMLU dataset on *general knowledge*, translated into 14 languages by human annotators. LLMs are prompted with a question and four candidate answers, and have to generate one option from $\{A, B, C, D\}$. In XCSQA, questions are also multi-choice (5 options, 16 languages), but focus on *commonsense reasoning*. By contrast, BMLAMA (Qi et al., 2023) includes parallel sentence prefixes and a varying number of possible parallel completions (e.g. "The capital of Italy is __", "{Rome/Paris/...}"), evaluating LLMs' *factual associations* in 17 languages. More detailed statistics and examples are provided in App. H.

**Evaluation Metrics.** We measure **crosslingual consistency** via the RankC metric (Qi et al., 2023), which considers the likelihood distribution over all candidate completions.[2] Besides, we evaluate **answer accuracy** following the LM-Evaluation-Harness framework[3] (Gao et al., 2024), where the candidate completion with the highest model likelihood is selected and compared to the gold answer.

## 6 RESULTS AND ANALYSIS

### 6.1 COMPARISON WITH PREVIOUS METHODS

We compare DCO with three representative methods: SFT, DPO, and CALM. Among these, SFT and DPO use *ground-truth labels* as the training target or the 'preferred' completion. To investigate the complementarity between DCO and DPO, we also evaluate a hybrid approach where the model is first trained with DPO and then refined with DCO using the *same* instances used for DPO. For a fair comparison, we follow the setup of Wang et al. (2025b), where each LLM is aligned across $N$ languages jointly in a single post-training process. In this setup, we use $\beta = 1$ and $\gamma_{ij} = 1$ for all $i, j \in \{1, \ldots, N\}$ with $N = 12$ on the general knowledge dataset MMMLU.[4]

| Method | Qwen2.5-14B | | | Gemma3-12B-pt | | | Qwen3-14B | | | Aya-Expanse-8B | | | Llama3.1-8B | | |
|---|---|---|---|---|---|---|---|---|---|---|---|---|---|---|---|
| | **CLC$_{\mathbf{all}}$** | **A$_{\mathbf{EN}}$** | **A$_{\mathbf{\neg EN}}$** | **CLC$_{\mathbf{all}}$** | **A$_{\mathbf{EN}}$** | **A$_{\mathbf{\neg EN}}$** | **CLC$_{\mathbf{all}}$** | **A$_{\mathbf{EN}}$** | **A$_{\mathbf{\neg EN}}$** | **CLC$_{\mathbf{all}}$** | **A$_{\mathbf{EN}}$** | **A$_{\mathbf{\neg EN}}$** | **CLC$_{\mathbf{all}}$** | **A$_{\mathbf{EN}}$** | **A$_{\mathbf{\neg EN}}$** |
| Base | 68.6±0.02 | 72.5±0.4 | 58.1±0.2 | 73.6±0.02 | 70.1±0.4 | 62.3±0.2 | 76.1±0.02 | 76.6±0.4 | 67.3±0.1 | 72.2±0.02 | 59.8±0.5 | 52.9±0.2 | 60.9±0.02 | 57.3±0.5 | 45.8±0.2 |
| + SFT* | +0.6±0.02 | +1.5±0.4 | +6.7±0.2 | +0.6±0.02 | +0.7±0.4 | +1.6±0.2 | −0.2±0.02 | +0.1±0.5 | +0.5±0.1 | +3.5±0.02 | +0.7±0.5 | +0.5±0.2 | +4.3±0.02 | +6.7±0.5 | +5.9±0.2 |
| + DPO* | +12.3±0.02 | +7.8±0.4 | +13.9±0.1 | +6.5±0.02 | +1.8±0.4 | +3.4±0.1 | +3.0±0.02 | +2.7±0.5 | +4.2±0.1 | +1.3±0.02 | +2.5±0.5 | +2.5±0.2 | +10.1±0.02 | +8.0±0.4 | +8.8±0.2 |
| + DCO* | +13.1±0.02 | +7.6±0.4 | +13.5±0.1 | +10.2±0.02 | +1.2±0.4 | +2.9±0.1 | +4.4±0.02 | +2.8±0.5 | +4.3±0.1 | +3.1±0.02 | +2.7±0.5 | +2.6±0.2 | +13.8±0.02 | +7.3±0.4 | +8.9±0.2 |
| + CALM | +4.2±0.02 | +0.0±0.4 | +4.1±0.2 | +3.0±0.02 | −0.4±0.4 | −0.0±0.2 | +0.3±0.02 | −2.1±0.5 | −1.1±0.1 | +1.4±0.02 | −2.2±0.5 | −2.1±0.2 | +3.0±0.02 | −2.2±0.5 | −5.0±0.2 |
| + DCO | +10.6±0.02 | +4.0±0.4 | +9.6±0.2 | +6.5±0.02 | +0.9±0.4 | +2.5±0.2 | +2.7±0.02 | +0.4±0.5 | +1.3±0.1 | +5.3±0.02 | +0.5±0.5 | +0.5±0.2 | +9.4±0.02 | +7.5±0.5 | +7.6±0.2 |

Table 1: Comparison with previous methods in the joint training setup, on the MMMLU dataset. **CLC$_{\mathbf{all}}$**: average crosslingual consistency (measured by RankC) between all language pairs; **A$_{\mathbf{EN}}$/A$_{\mathbf{\neg EN}}$**: average accuracy on English/non-English instances. We report mean scores with their standard errors. Methods with * are trained on ground-truth answers.

As shown in Tab. 1, the three baselines have distinct behaviors. *SFT*, trained on instances with gold labels, produces modest gains: it slightly improves **CLC$_{\mathbf{all}}$** and accuracies (e.g., `Llama3.1-8B`), yet on stronger models such as Qwen3-14B, its effect on **CLC$_{\mathbf{all}}$** is negligible or even slightly negative, indicating that simple SFT is not a reliable mechanism for enhancing consistency. In contrast, *DPO* yields larger gains in **CLC$_{\mathbf{all}}$**, A$_{\mathbf{EN}}$, and A$_{\mathbf{\neg EN}}$ across all LLMs.

---

[2] See App. I for detailed definitions of the function.

[3] https://github.com/EleutherAI/lm-evaluation-harness

[4] We exclude Bengali as its prompt length exceeds the memory capacity of four A100 GPUs, and Swahili and Yoruba because most of our LLMs perform at a random-guess level in these languages. See §6.4 for targeted experiments on the low-resource languages, which demonstrate promising results with adjusted $\gamma$ values.

Turning to label-free approaches, CALM applies DPO to encourage a preference for majority answers across different languages. This design requires more than two languages, limiting applicability in bilingual settings; moreover, including multiple low-resource languages can make majority voting unreliable. Empirically, this limitation is evident in our results: $\mathbf{CLC_{all}}$ is slightly improved, English and non-English accuracy fluctuates without obvious increments, confirming its sensitivity to noisy majority voting when low-resource languages are included. By contrast, DCO yields consistently higher $\mathbf{CLC_{all}}$ on all tested models while preserving accuracy or even improving it in many cases. Notably, on some models (e.g., `Aya-Expanse-8B`) DCO even surpasses DPO on $\mathbf{CLC_{all}}$, and on the rest it nearly matches DPO despite using no gold labels. We provide detailed CLC results for all language pairs in App. L, showing that DCO not only improves CLC for typologically similar languages, but also for distant pairs such as Korean-French and Arabic–Chinese.

Finally, combining gold-label preference learning with consistency optimization (DPO+DCO) yields optimal results: applying DCO as a post-step to a DPO-trained model consistently achieves the highest $\mathbf{CLC_{all}}$ across all language models. Accuracy remains comparable to DPO, with minor trade-offs in English for some models, and slight gains in non-English languages for others. Taken together, these results highlight DCO as the most versatile and practical option. It offers a robust path to crosslingual consistency while preserving (and often improving) task accuracy, and it can also serve as an effective post-step that further benefits models already trained with DPO.

## 6.2 BILINGUAL IMPROVEMENTS

The experiments in §6.1 target *joint* consistency improvements across many languages, a setting aligned with large multilingual foundation models. In practical scenarios, however, developers may be interested in aligning knowledge between English and a specific local language, or between a small set of regional languages. To assess this use case, we instantiate a *bilingual* version of DCO that aligns English with one non-English language, and we extend the evaluation beyond MMMLU (general knowledge) to XCSQA (commonsense reasoning) and BMLAMA (factual association). Tab. 2 reports CLC and average answer accuracy on English and non-English. For space reasons, we present the largest model in each family (see App. M for full results on nine LLMs).

| Model | MMMLU | | | XCSQA | | | BMLAMA | | |
|---|---|---|---|---|---|---|---|---|---|
| | $\mathbf{CLC_{all}}$ | $\mathbf{A_{EN}}$ | $\mathbf{A_{\neg EN}}$ | $\mathbf{CLC_{all}}$ | $\mathbf{A_{EN}}$ | $\mathbf{A_{\neg EN}}$ | $\mathbf{CLC_{all}}$ | $\mathbf{A_{EN}}$ | $\mathbf{A_{\neg EN}}$ |
| `Qwen2.5-14B` | $68.6 \pm 0.02$ | $72.5 \pm 0.4$ | $58.1 \pm 0.2$ | $64.6 \pm 0.02$ | $87.0 \pm 2.5$ | $56.9 \pm 0.9$ | $41.9 \pm 0.03$ | $62.7 \pm 3.2$ | $38.6 \pm 1.0$ |
| + DCO | $+12.6 \pm 0.02$ | $+1.6 \pm 0.4$ | $+8.5 \pm 0.2$ | $+6.8 \pm 0.02$ | $-2.5 \pm 2.5$ | $+4.7 \pm 0.9$ | $+15.4 \pm 0.02$ | $+6.3 \pm 3.2$ | $+14.2 \pm 1.0$ |
| `Gemma3-12B-pt` | $73.6 \pm 0.02$ | $70.1 \pm 0.4$ | $62.3 \pm 0.2$ | $58.3 \pm 0.02$ | $66.0 \pm 3.4$ | $47.2 \pm 0.9$ | $42.2 \pm 0.03$ | $68.3 \pm 3.2$ | $38.3 \pm 1.0$ |
| + DCO | $+7.2 \pm 0.02$ | $-0.9 \pm 0.4$ | $+1.4 \pm 0.2$ | $+4.6 \pm 0.02$ | $+0.1 \pm 3.4$ | $+3.6 \pm 0.9$ | $+16.7 \pm 0.02$ | $+1.6 \pm 3.2$ | $+17.0 \pm 1.0$ |
| `Qwen3-14B` | $76.1 \pm 0.02$ | $76.6 \pm 0.4$ | $69.0 \pm 0.1$ | $61.9 \pm 0.02$ | $77.6 \pm 3.0$ | $54.0 \pm 0.9$ | $38.9 \pm 0.02$ | $58.4 \pm 3.2$ | $36.4 \pm 1.0$ |
| + DCO | $+4.8 \pm 0.02$ | $+0.1 \pm 0.4$ | $+1.7 \pm 0.1$ | $+7.1 \pm 0.02$ | $+1.1 \pm 3.0$ | $+3.8 \pm 0.9$ | $+16.2 \pm 0.02$ | $+8.1 \pm 3.2$ | $+14.5 \pm 1.0$ |
| `Aya-Expanse-8B` | $72.2 \pm 0.02$ | $59.8 \pm 0.4$ | $52.9 \pm 0.1$ | $62.6 \pm 0.02$ | $78.0 \pm 3.0$ | $54.4 \pm 0.9$ | $41.9 \pm 0.02$ | $67.0 \pm 3.2$ | $37.8 \pm 1.0$ |
| + DCO | $+5.3 \pm 0.02$ | $+0.5 \pm 0.4$ | $+0.5 \pm 0.1$ | $+6.4 \pm 0.02$ | $+0.6 \pm 3.0$ | $+3.7 \pm 0.9$ | $+12.3 \pm 0.02$ | $+1.4 \pm 3.2$ | $+12.2 \pm 1.0$ |
| `Llama3.1-8B` | $60.9 \pm 0.02$ | $57.3 \pm 0.5$ | $45.8 \pm 0.2$ | $60.2 \pm 0.02$ | $67.5 \pm 3.3$ | $47.7 \pm 0.9$ | $40.9 \pm 0.03$ | $61.2 \pm 3.5$ | $35.8 \pm 1.0$ |
| + DCO | $+12.1 \pm 0.02$ | $+0.7 \pm 0.5$ | $+3.0 \pm 0.2$ | $+9.1 \pm 0.02$ | $+0.2 \pm 3.3$ | $+1.3 \pm 0.9$ | $+15.7 \pm 0.02$ | $+7.2 \pm 3.2$ | $+17.6 \pm 1.0$ |

Table 2: Consistency with English and average accuracy of all English and non-English pairs on MMMLU, XCSQA, and BMLAMA, with standard deviations. See App. M for full results on nine LLMs.

Overall, DCO substantially improves both $\mathbf{CLC_{all}}$ and accuracy across datasets. On MMMLU, $\mathbf{CLC_{all}}$ increases by +4.79 to +12.60 across all models, with concurrent gains in the accuracy of non-English $A_{\neg EN}$ (+0.46 to +8.49) and remains largely stable in English accuracy. On XCSQA, $\mathbf{CLC_{all}}$ improves by +4.61 to +9.10, with smaller changes in English accuracy (–2.53 to +1.07, with the single notable dip on `Qwen2.5-14B`), while non-English accuracy increases consistently (+1.27 to +4.67). The largest gains appear on BMLAMA, where $\mathbf{CLC_{all}}$ improves by +12.29 to +16.65 and both English accuracy (+1.43 to +8.07) and non-English accuracy (+12.16 to +17.62) rise markedly. We hypothesize that BMLAMA benefits more from DCO because outputs are concrete factual entities rather than abstract option labels, making distributional alignment across languages more direct.

## 6.3 OUT-OF-DOMAIN GENERALIZABILITY

To assess whether the benefits of our method extend beyond the specific domain on which the model was post-trained, we conduct a controlled experiment using `Qwen2.5-14B`: DCO is performed on

a single subject within MMMLU (namely, *high school microeconomics*) and evaluated on various other subjects from the same dataset. For easier interpretation of the results and for managing computational costs, we keep the bilingual DCO setup in the rest of our experiments.

| Method | Anatomy | | | Medical Genetics | | | High School Maths | | | College Maths | | |
|---|---|---|---|---|---|---|---|---|---|---|---|---|
| | $CLC_{all}$ | $A_{EN}$ | $A_{\neg EN}$ | $CLC_{all}$ | $A_{EN}$ | $A_{\neg EN}$ | $CLC_{all}$ | $A_{EN}$ | $A_{\neg EN}$ | $CLC_{all}$ | $A_{EN}$ | $A_{\neg EN}$ |
| Base | 59.49 | 68.89 | 46.30 | 70.38 | 82.00 | 63.79 | 67.83 | 53.70 | 43.23 | 64.25 | 57.00 | 37.50 |
| + DCO | +10.94 | +1.80 | +3.76 | +10.33 | +5.43 | +7.00 | +11.27 | +2.38 | +6.80 | +11.36 | -0.36 | +12.36 |

Table 3: Cross-domain performance on `Qwen2.5-14B`. The model is trained with DCO on *high school microeconomics* (390 questions) and tested on distinct domains on MMMLU. Detailed results by language, and for more test domains, are shown in App. M.

Tab. 3 shows strong out-of-domain transfer from a single training subject. DCO increases $CLC_{all}$ by about 11% on average across all target domains, indicating that CLC is enhanced beyond the specific post-training domain. Non-English accuracy also significantly improves, with the largest gain on *college mathematics* (+12.36), reflecting effective knowledge transfer from English to less resourced languages, even without explicit accuracy supervision. English accuracy is largely preserved, with only a negligible decrease of 0.36 on *college mathematics*, showing that DCO does not overfit to the training subject or degrade the model's primary language competence. Taken together, these results support the potential of DCO as a practical tool for real-world deployments where labeled data are scarce and the target application domain may differ from that of the available training data.

### 6.4 EFFECT OF DIRECTION CONTROLLING PARAMETERS

**Setup.** We study how the parameters $\gamma_1, \gamma_2$ of DCO control transfer specifically between English (EN) and low-resource languages. Specifically, we select Swahili (SW) and Yoruba (YO), which have the lowest baseline accuracy on MMMLU, and vary the values of $\gamma_1, \gamma_2$ in three regimes[5]: **Default** ($\gamma_1=1, \gamma_2=1$), **SW/YO Stable** ($\gamma_1=0.1, \gamma_2=10$, strong transfer from SW/YO to EN), and **EN Stable** ($\gamma_1=10, \gamma_2=0.1$, strong transfer from EN to SW/YO)). All other settings remain the same as in §6.2.

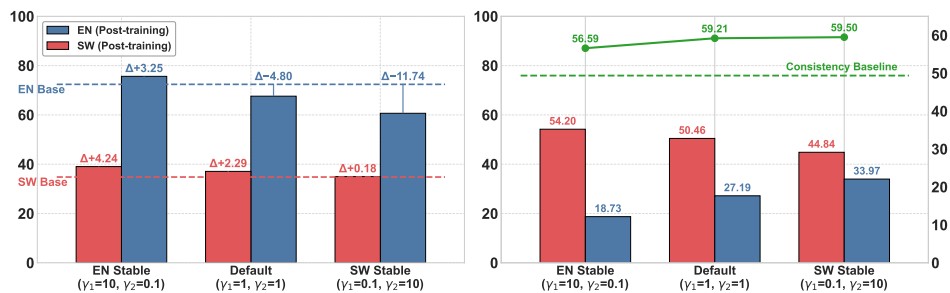

Figure 2: Left: Answer accuracy after performing DCO on English-Swahili. Right: Proportion of questions for which the LLM's response changes after DCO, with CLC values marked in green.

**Results.** We present results of DCO between EN and SW in Fig. 2 and leave the results between EN and YO to App. J where similar trends are observed. With default weighting, EN accuracy declines by $-4.80$ points while SW gains $+2.29$. As we expected, the SW-stable weighting (large $\gamma_2$) severely hurts EN accuracy ($-11.74$) while barely helping SW ($+0.18$). Finally, the EN-stable weighting (large $\gamma_1$) yields a Pareto improvement: the accuracy of SW increases by $+4.24$ while the accuracy in EN remains less changed (smaller $\Delta$); in this case, fortunately, it even slightly increases. By contrast, CLC improves in all weighting schemes. The baseline $CLC_{all}$ of 49.37 rises to 59.21, 59.50, 56.59 under $(\gamma_1:\gamma_2) = (1:1), (0.1:10), (10:0.1)$ respectively, showing that DCO reliably optimizes consistency across different direction weights. Notably, the larger $CLC_{all}$ boost from the default weighting comes with a substantial EN accuracy drop, whereas the EN-stable setup achieves a more favorable balance: that is, slightly smaller yet still significant gains in $CLC_{all}$ but stable or improved EN accuracy.

---

[5]Note that we keep $\gamma_1 \gamma_2 = \beta^2 = 1$ as discussed in Lemma 1.

**Ratio of changed answers after DCO.** To better understand the effects of $(\gamma_1, \gamma_2)$, we measure the proportion of questions for which the model answer changes after DCO. As shown in Fig. 2 (right), the low-resource side exhibits far more updates than EN, and the direction weights control which side is allowed to move. The EN-stable setting changes only 18.73% of EN answers but 54.20% of SW. In default weighting, EN changes increase and SW changes decrease. In the SW-stable setting, the burden shifts to EN (33.97% EN vs. 44.84% SW). A similar trend holds for English-Yoruba (see App. J). These patterns align with the accuracy/consistency results: EN-anchored regimes keep the high-resource channel stable while DCO primarily revises the low-resource outputs, thereby yielding improved CLC and higher non-EN accuracy. On the other hand, low-resource stable setups induce unnecessary EN churn without sufficient benefits. However, this should not be misinterpreted as '*always set a large $\gamma$ for* EN'. We provide weighting guidance for real-world applications in App. K.

## 6.5 DISCUSSION

**Where do the accuracy gains come from?** In general, when a language model performs poorly on a task, it tends to have a *high-entropy distribution* over the candidate answers. In contrast, a low-entropy one that is skewed toward an incorrect answer is less common. Thus, the experts in $\pi^\star(\boldsymbol{y} \mid \boldsymbol{x}) = \frac{1}{Z} \prod_j \left( \pi_{\text{REF}}\big(\tau^j(\boldsymbol{y}) \mid \tau^j(\boldsymbol{x})\big) \right)^{w_j}$ are complementary to each other. Specifically, a low-entropy expert only contributes minimally to the final distribution, allowing the ensemble to rely more on high-confidence predictions from other experts. As a case study, we verify this assumption using Qwen-2.5-14B on the MMMLU dataset, where the average entropy of the answer distribution on the questions that are correctly answered is $0.41 \pm 0.41$, while for incorrectly answered questions, it is significantly higher at $0.98 \pm 0.33$. The accuracy of the theoretical optimal policy $\pi^\star$ on MMMLU using Qwen-2.5-14B is 77.0, surpassing the accuracy of the base policy in individual languages.

**Beyond crosslingual consistency.** While this work focuses on *crosslingual* knowledge consistency, the training objective of DCO is not limited to this task and can naturally be extended to other forms of consistency. For instance, by aligning the output distributions over candidate answers for paraphrased prompts, the model could be encouraged to respond consistently regardless of surface variations. Exploring such extensions is an interesting direction for future work.

## 7 CONCLUSION

This paper proposes a novel structured reward function to improve crosslingual consistency in multilingual LLMs and introduces an efficient method, direct consistency optimization (DCO).

Through comprehensive experiments, we demonstrate that DCO consistently improves CLC across a variety of models and datasets. Compared to existing methods, DCO delivers robust performance gains and complements DPO when gold labels are available, producing the strongest overall knowledge alignment in a joint $N$-languages training setting. In bilingual settings, DCO also enhances CLC effectively, raising accuracy in non-English languages while maintaining accuracy in English. We further show the generalizability of DCO across domains, with gains observed even when testing on subjects that differ from the training ones. The analysis of direction-controlling weights demonstrates how practitioners can steer alignment toward specific languages according to deployment requirements.

Looking ahead, we believe the structured reward underlying DCO has potential for application beyond crosslingual knowledge consistency, for example, improving self-consistency across paraphrases or consistency across modalities. As a computationally efficient algorithm, DCO provides a practical path toward building powerful multilingual LLMs that are not only accurate but also reliable and equitable across languages.

## REPRODUCIBILITY STATEMENT

We provide detailed theoretical proofs of Lemmas 1 and 2 in Apps. D.1 and D.2. Implementation details and training configurations are given in App. G. Datasets are contained in the anonymous supplementary material.

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

## A    USAGE OF LLMS

LLM tools were used occasionally to improve writing clarity. They did not contribute to the conceptual development, experimental design, or analysis. The authors reviewed and edited all assisted text, and the final manuscript is *entirely* author-written.

## B    BACKGROUND: MODELING HUMAN PREFERENCES WITH BRADLEY-TERRY MODEL

An important component for post-training LLMs is reward models that align with human preferences (Ziegler et al., 2020; Wu et al., 2021; Ouyang et al., 2022). To construct a human feedback dataset $\mathcal{D}_{\mathrm{HF}}$,

humans are shown two (or more) responses to a prompt $\boldsymbol{x}$ and are asked to select the response they prefer. $\mathcal{D}_{\text{HF}}$ is denoted as a collection of triples $(\boldsymbol{x}, \boldsymbol{y}_w, \boldsymbol{y}_l)$, where $\boldsymbol{y}_w$ is preferred over $\boldsymbol{y}_l$ by a human.

To train a reward function $r_\phi$ on $\mathcal{D}_{\text{HF}}$, it is common to assume that human preference can be modeled by a Bradley–Terry model (Bradley & Terry, 1952),

$$\mathbb{P}_{r_\phi}(\boldsymbol{y}_w \succ \boldsymbol{y}_l) \stackrel{\text{def}}{=} \sigma\big(r_\phi(\boldsymbol{x}, \boldsymbol{y}_w) - r_\phi(\boldsymbol{x}, \boldsymbol{y}_l)\big), \tag{13}$$

where $\sigma(x) \stackrel{\text{def}}{=} \frac{1}{1+\exp(-x)}$ is the sigmoid function. The reward model $r_\phi$, parameterized by $\phi$, is trained to minimize the following negative log-likelihood loss:

$$\mathcal{L}^r(\phi) = - \mathbb{E}_{(\boldsymbol{x}, \boldsymbol{y}_w, \boldsymbol{y}_l) \sim \mathcal{D}_{\text{HF}}} \Big[ \log \mathbb{P}_{r_\phi}(\boldsymbol{y}_w \succ \boldsymbol{y}_l) \Big]. \tag{14}$$

Intuitively, the reward function should assign higher rewards to the responses that are preferred by humans. Then, the reward function is then plugged into Eq. (1) for policy optimization.

## C   DISCUSSION ON THE DEFINITION OF CROSSLINGUAL CONSISTENCY

One might attempt to use a stricter definition of consistency, such as requiring exact probability matches $\pi^\star(\boldsymbol{y}_w^1 \mid \boldsymbol{x}^1) = \pi^\star(\boldsymbol{y}_w^2 \mid \boldsymbol{x}^2)$. Previous work has shown that even semantically identical sentences in different languages can have likelihoods that differ significantly due to lexical, semantic, and tokenization differences (Lesci et al., 2025). For instance, in `Gemma3-12B-it`, using a temperature of 1, $\pi(\text{" Paris"} \mid \text{"The capital of France is"}) = 0.8991$, while $\pi(\text{" Paris"} \mid$ "Die Hauptstadt Frankreichs ist") $= 0.1283$. Thus, enforcing exact probability matches would ignore these inherent differences and lead to suboptimal alignment. Instead, we adopt a softer order-based consistency constraint.

## D   PROOFS

### D.1   PROOF OF LEMMA 1

**Lemma 1.** *If $\gamma_1 \gamma_2 = \beta^2$, the optimal policy $\pi^\star$ defined by Eq. (8) is consistent across $L_1$ and $L_2$.*

*Proof.* Assume $\gamma_1 \gamma_2 = \beta^2$. Raising both sides of Eq. (9a) to the power of $\frac{\beta}{\gamma_1}$, we obtain:

$$\big(\pi^\star(\boldsymbol{y}^1 \mid \boldsymbol{x}^1)\big)^{\beta/\gamma_1} = \frac{1}{Z^{\beta/\gamma_1}(\boldsymbol{x}^1)} \big(\pi_{\text{REF}}(\boldsymbol{y}^1 \mid \boldsymbol{x}^1)\big)^{\beta/\gamma_1} \pi_{\text{REF}}(\tau^2(\boldsymbol{y}^1) \mid \tau^2(\boldsymbol{x}^1))$$

$$= \frac{1}{Z^{\beta/\gamma_1}(\boldsymbol{x}^1)} \underbrace{\big(\pi_{\text{REF}}(\boldsymbol{y}^1 \mid \boldsymbol{x}^1)\big)^{\gamma_2/\beta} \pi_{\text{REF}}(\tau^2(\boldsymbol{y}^1) \mid \tau^2(\boldsymbol{x}^1))}_{= Z(\tau^2(\boldsymbol{x}^1)) \pi^\star(\tau^2(\boldsymbol{y}^1) \mid \tau^2(\boldsymbol{x}^1)) \quad \text{(Eq. (9b))}} \qquad (\gamma_1 \gamma_2 = \beta^2)$$

$$\equiv \frac{Z(\tau^2(\boldsymbol{x}^1))}{Z^{\beta/\gamma_1}(\boldsymbol{x}^1)} \pi^\star(\tau^2(\boldsymbol{y}^1) \mid \tau^2(\boldsymbol{x}^1)).$$

Note that the term $\frac{Z(\tau^2(\boldsymbol{x}^1))}{Z^{\beta/\gamma_1}(\boldsymbol{x}^1)}$ is positive and independent of $\boldsymbol{y}^1$. Since the function $f(x) = cx^{\beta/\gamma_1}$ increases monotonically in $x$ for $\beta/\gamma_1 > 0, c > 0$, we have $\pi^\star(\boldsymbol{y}_w^1 \mid \boldsymbol{x}^1) \geq \pi^\star(\boldsymbol{y}_l^1 \mid \boldsymbol{x}^1) \iff \pi^\star(\tau^2(\boldsymbol{y}_w^1) \mid \tau^2(\boldsymbol{x}^1)) \geq \pi^\star(\tau^2(\boldsymbol{y}_l^1) \mid \tau^2(\boldsymbol{x}^1))$, for all $\boldsymbol{y}_w^1, \boldsymbol{y}_l^1$. Thus, $\pi^\star$ is consistent across $L_1$ and $L_2$. ∎

### D.2   PROOF OF LEMMA 2

**Lemma 2.** *When Eq. (11) is minimized, the reward function $\widehat{r}_{\boldsymbol{\theta}}$ will converge to*

$$\widehat{r}_{\boldsymbol{\theta}}^\star(\boldsymbol{x}, \boldsymbol{y}) = \begin{cases} \gamma_1 \log \pi_{\text{REF}}(\tau^2(\boldsymbol{y}) \mid \tau^2(\boldsymbol{x})) + c(\boldsymbol{x}) & \text{if } \boldsymbol{x}, \boldsymbol{y} \in L_1, \\ \gamma_2 \log \pi_{\text{REF}}(\tau^1(\boldsymbol{y}) \mid \tau^1(\boldsymbol{x})) + c(\boldsymbol{x}) & \text{if } \boldsymbol{x}, \boldsymbol{y} \in L_2, \end{cases} \tag{12}$$

*where $c(\boldsymbol{x})$ is a function independent of $\boldsymbol{y}$.*

*Proof.* Following Rafailov et al. (Definition 1, 2023), two reward functions $r(\boldsymbol{x}, \boldsymbol{y})$ and $r'(\boldsymbol{x}, \boldsymbol{y})$ are **equivalent** if $r(\boldsymbol{x}, \boldsymbol{y}) - r'(\boldsymbol{x}, \boldsymbol{y}) = f(\boldsymbol{x})$ for some function $f$. Our goal is to show that minimizing $L(\boldsymbol{\theta})$ recovers a reward function equivalent to $r_{\text{ALIGN}}$ (Eq. (6)).

First, note that $L(\boldsymbol{\theta}) \geq 0$ in Eq. (11), due to the non-negativity of the absolute value function. Substituting Eq. (12) into Eq. (11), we find that $L(\boldsymbol{\theta}) = 0$, which implies that $\widehat{r}_{\boldsymbol{\theta}}{}^{\star}$ minimizes the loss.

Furthermore, since $\widehat{r}_{\boldsymbol{\theta}}{}^{\star}$ satisfies the structure of Eq. (12), it is equivalent to $r_{\text{ALIGN}}$ up to an additive term $c(\boldsymbol{x})$, which does not affect the policy optimization. ∎

# E  GENERALIZATION TO $N$ LANGUAGES

We generalize §4 to aligning $N$ languages. The reward function for alignment is generalized to

$$r_{\text{ALIGN}}(\boldsymbol{x}, \boldsymbol{y}) = \begin{cases} \sum_{j=1}^{N} \gamma_{ij} \log \pi_{\text{REF}}(\tau^j(\boldsymbol{y}) \mid \tau^j(\boldsymbol{x})) & \text{when } \boldsymbol{x}, \boldsymbol{y} \in L_i, \\ 0 & \text{otherwise.} \end{cases} \tag{15}$$

We set $\gamma_{ii} = 0$ for $i \in \{1, 2, \cdots, N\}$. Thus, there are $(N^2 - N)$ hyperparameters in total.

The optimal policy is

$$\pi^{\star}(\boldsymbol{y}^i \mid \boldsymbol{x}^i) = \frac{1}{Z(\boldsymbol{x}^i)} \pi_{\text{REF}}(\boldsymbol{y}^i \mid \boldsymbol{x}^i) \prod_{j \in \{1, 2, \cdots, N\} \setminus \{i\}} \left( \pi_{\text{REF}}(\tau^j(\boldsymbol{y}^i) \mid \tau^j(\boldsymbol{x}^i)) \right)^{\gamma_j/\beta}. \tag{16}$$

We define the following matrix

$$\Gamma \overset{\text{def}}{=} \begin{pmatrix} 1 & \gamma_{12}/\beta & \cdots & \gamma_{1N}/\beta \\ \gamma_{21}/\beta & 1 & \cdots & \gamma_{2N}/\beta \\ \vdots & \vdots & \vdots & \vdots \\ \gamma_{N1}/\beta & \gamma_{22}/\beta & \cdots & 1 \end{pmatrix} \tag{17}$$

We give a sufficient condition on $\Gamma$ that leads to consistent policies.

**Lemma 3.** *When* $\text{rank}(\Gamma) = 1$, $\pi^{\star}$ *is consistent across* $L_1, \cdots, L_N$.

*Proof.* When $\text{rank}(\Gamma) = 1$, each row of $\Gamma$ can be represented as a multiple of the first row. I.e., there exists $(N-1)$ numbers $k_2, \cdots, k_N$ such that

$$\begin{pmatrix} 1 & \gamma_{12}/\beta & \cdots & \gamma_{1N}/\beta \end{pmatrix} = k_i \begin{pmatrix} \gamma_{i2}/\beta & \gamma_{i2}/\beta & \cdots & \gamma_{iN}/\beta \end{pmatrix}. \tag{18}$$

Then, $\pi^{\star}(\boldsymbol{y}^1 \mid \boldsymbol{x}^1) = \frac{(Z(\boldsymbol{x}^i))^{k_i}}{Z(\boldsymbol{x}^1)} \left( \pi^{\star}(\boldsymbol{y}^i \mid \boldsymbol{x}^i) \right)^{k_i}$, which indicates $\pi^{\star}(\boldsymbol{y}^i \mid \boldsymbol{x}^i)$ is consistent with $\pi^{\star}(\boldsymbol{y}^1 \mid \boldsymbol{x}^1)$ for $i \in \{2, \cdots, N\}$. Therefore, every pair of languages $L_i, L_j$ is consistent. ∎

# F  COMPARISON WITH THE DPO OBJECTIVE

The DPO objective relies on a labeled dataset of preferences, aiming to train the Bradley–Terry model $\mathbb{P}_{\widehat{r}_{\boldsymbol{\theta}}}(\boldsymbol{y}_w \succ \boldsymbol{y}_l) = \sigma(\beta(\widehat{r}_{\boldsymbol{\theta}}(\boldsymbol{x}, \boldsymbol{y}_w) - \widehat{r}_{\boldsymbol{\theta}}(\boldsymbol{x}, \boldsymbol{y}_l)))$ to match ground truth preference labels. The optimal $\widehat{r}_{\boldsymbol{\theta}}$ in this case can take *unbounded* values and lacks a closed-form expression. By contrast, in $\mathcal{D}_{\parallel}$, the response pairs $\boldsymbol{y}_w^1, \boldsymbol{y}_l^1$ are randomly paired, meaning $\boldsymbol{y}_w^1$ is not necessarily the better response carrying the gold answer, which benefits real-world applications. Minimizing $L(\boldsymbol{\theta})$ ensures that the Bradley–Terry model $\mathbb{P}_{\widehat{r}_{\boldsymbol{\theta}}}$ matches the distribution $\sigma\left( \log \frac{\pi_{\text{REF}}(\boldsymbol{y}_w^2 \mid \boldsymbol{x}^2)}{\pi_{\text{REF}}(\boldsymbol{y}_l^2 \mid \boldsymbol{x}^2)} \right)$ exactly.

# G  IMPLEMENTATION DETAILS

In our experiments, we set $\beta = 1$, $\gamma_1 = \gamma_2 = 1$. For all models, we use the AdamW optimizer with a learning rate of $1e^{-5}$, with an exception of $1e^{-6}$ for Gemma models on XCSQA to avoid overfit, weight decay of 0, $\beta_1 = 0.9, \beta_2 = 0.999, \varepsilon = 1e^{-8}$. All models are trained on four A100 GPUs of 40GB memory. For SFT, DPO, and CALM, the learning rate is set to $5e^{-7}$, $5e^{-6}$, and $5e^{-6}$, respectively.

# H  DATASET DETAILS

Three datasets with parallel queries and candidate completions are used in our experiments. Here we list the detailed statistics in Tab. 4 and examples in Tab. 5.

| Dataset | Knowledge Type | #Langs | Paralleled? | | #Train | #Test | Answer Format |
|---------|----------------|--------|-------|-----------|--------|-------|---------------|
| | | | Query | Candidate | | | |
| MMMLU | General Knowledge | 12(+2) | ✓ | ✓ | 5000 | 9042 | A/B/C/D |
| XCSQA | Commonsense Reasoning | 16 | ✓ | ✓ | 800 | 200 | A/B/C/D/E |
| BMLAMA | Factual Association | 17 | ✓ | ✓ | 5000 | 1792 | Words |

Table 4: Statistics of MMMLU, XCSQA and BMLAMA datasets.

| Dataset | Query | Candidates | Gold Answer |
|---------|-------|------------|-------------|
| MMMLU | Which cells in the blood do not have a nucleus?
A. Lymphocyte
B. Monocyte
C. Erythrocyte
D. Basophil | [A, B, C, D] | C |
| XCSQA | What might lock after someone drives in?
A. gate
B. doorknob
C. mouths
D. entrance
E. front door | [A, B, C, D, E] | A |
| BMLAMA | Berlin is the capital of | [Greenland, Piedmont, Oman, Venezuela, Fiji, Latvia, Taiwan, Norway, Romania, Germany] | Germany |

Table 5: Examples of instances in MMMLU, XCSQA and BMLAMA.

**MMMLU.**   MMMLU is a multilingual extension of the MMLU dataset (Hendrycks et al., 2021) on general knowledge. The LLMs with the question and candidate answers. The LLMs are expected to give an answer from $\{A, B, C, D\}$. Each question and its candidate answers are given in 15 languages. In our experiments, we exclude Bengali due to the GPU constraint, and Swahili and Yoruba due to their low accuracy. Nonetheless, we conduct extra experiments and in-depth analysis on these two languages (i.e., brackets in Tab. 4) in §6.4.

**XCSQA.**   Similar to MMMLU, the questions are multi-choice commonsense reasoning tasks over 16 languages. The answer space is also a set of capital letters: $\{A, B, C, D, E\}$.

**BMLAMA.**   The BMLAMA dataset Qi et al. (2023) specifically focuses on factual associations, and the answer format is objective words rather than option letters, which promotes the best crosslingual knowledge alignment in our experiments. Parallel prompts and candidate answers are provided across 17 languages.

**Sampling training instances.**   Regarding MMMLU and BMLAMA, we randomly sample two pairs of parallel candidate completions per query in the training set, yielding 5000 instances for DCO. Regarding XCSQA, we repeat this sampling procedure seven times to construct a dataset of comparable size (800*7=5600) for DCO training.

# I  DETAILS OF RANKC: RANKING-BASED CROSSLINGUAL CONSISTENCY

*RankC* (Qi et al., 2023) is a ranking-based consistency metric for assessing crosslingual knowledge consistency independently of probing accuracy.

Given a parallel query set: $Q^1 = \{x_i^1\}_{i=1}^{|Q|}, \quad Q^2 = \{x_i^2\}_{i=1}^{|Q|}$, where each $x_i^1$ in $L_1$ corresponds to $x_i^2$ in $L_2$. For the $i$-th query, assuming there are $N_i$ candidate answers $\{c_{i,j}\}_{j=1}^{N_i}$, the model assigns a

likelihood to each of the candidates. Let the candidates in each language be sorted by descending likelihood: $c_{i,1}^1, c_{i,2}^1, \ldots, c_{i,N_i}^1$ (in $L_1$) and $c_{i,1}^2, c_{i,2}^2, \ldots, c_{i,N_i}^2$ (in $L_2$).

Then, the 'precision at $j$' (denoted $P@j$) is defined as the proportion of overlap among the top-$j$ candidates in both languages: $P@j = \frac{1}{j} \left| \{c_{i,1}^1, \ldots, c_{i,j}^1\} \cap \{c_{i,1}^2, \ldots, c_{i,j}^2\} \right|$. A ranking-based weight $w_j = \frac{\exp(N_i - j)}{\sum_{k=1}^{N_i} \exp(N_i - k)}$ is multiplied to each $P@j$, so that agreements at smaller $j$ (i.e., top of the list) are rewarded more. Given these, the consistency score for that query pair is: $\text{consist}(\boldsymbol{x}_i^1, \boldsymbol{x}_i^2) = \sum_{j=1}^{N_i} w_j \cdot P@j$.

Finally, the overall RankC between $L_1$ and $L_2$ is the average consistency score over all query pairs: $\text{RankC}(L_1, L_2) = \frac{1}{|Q|} \sum_{i=1}^{|Q|} \text{consist}(\boldsymbol{x}_i^1, \boldsymbol{x}_i^2)$.

## J    DIRECTION CONTROLLING RESULTS ON ENGLISH-YORUBA

Shown in Fig. 3, the original accuracy on EN and YO is more severe since it is an extremely low-resource language. The **default** weighting induces decreases in both languages: EN drops by $-15.94$ and YO also declines by $-1.18$. Making Yoruba 'stable' (**YO Stable**; $\gamma_1{=}0.1, \gamma_2{=}10$) exacerbates the problem, especially pushing EN down to $-19.97$ while still not helping Yoruba accuracy ($\Delta = -1.22$). In contrast, the **EN Stable** setup ($\gamma_1{=}10, \gamma_2{=}0.1$) delivers the best trade-off: EN accuracy remains less affected ($+2.98$) while that of Yoruba also improves by $+1.43$. Regarding CLC, for EN–YO, the baseline of 45.67 increases to 57.16, 55.40, and 51.66, demonstrating the effectiveness of DCO in crosslingual knowledge alignment.

As for the proportion of changed answers, similar to EN-SW, EN-stable minimizes EN updates as 19.00% while allowing substantial revisions on Yoruba as 59.87%. Default setup raises EN changes to 38.74%, and YO Stable setup further increases EN changes to 42.51%.

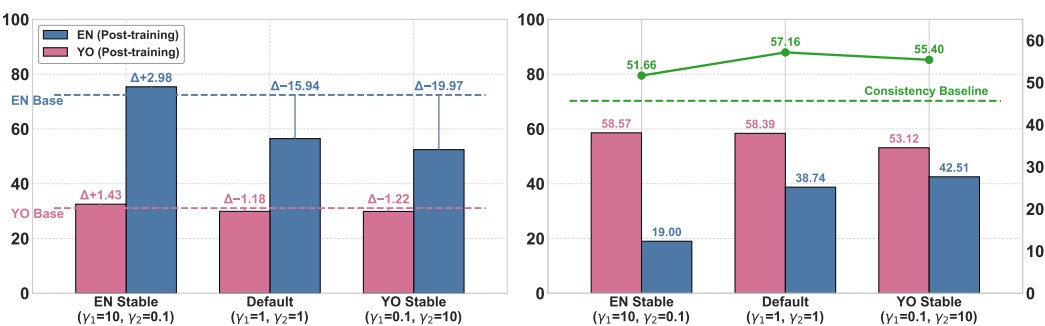

Figure 3: Left: Answer accuracy after performing DCO on English-Yoruba. Right: Proportion of questions for which the LLM's response changes after DCO, with CLC values marked in green.

## K    GUIDANCE OF DIRECTION CONTROLLING FOR REAL-WORLD APPLICATIONS

Direction–controlling hyperparameters matter: by adjusting $(\gamma_1, \gamma_2)$, we control the transfer strength for each side, thus deciding the direction of optimizing knowledge consistency. The low-resource languages Swahili and Yoruba were selected in our experiments for the best visibility of the effect; yet the results should not be misinterpreted as 'always set a large $\gamma$ for EN.' In fact, the principle is more general: anchor on the high-quality, high-priority language, which is EN in our study, but could be French or any other well-trained language, according to the specific requirements of the downstream LLM application. When both languages are of comparable quality, or when policy requires reciprocity, a symmetric schedule like 1:1 is expected to be optimal. In practice, $(\gamma_1, \gamma_2)$ can also be selected empirically against a small validation set. The 'ratio of changed answers' (i.e. Fig. 2 (right) & Fig. 3 (right)) is useful: if the intended stable language exhibits excessive changes, or the

language expected to shift shows little movement, increase $\gamma_1$, and by construction, $\gamma_2$ will decrease correspondingly ($\gamma_2 = 1/\gamma_1$).

Note that DCO is capable of improving CLC under any direction parameter setups, as demonstrated in Fig. 2 & Fig. 3. This empirical adjustment further yields gains in accuracy for both sides.

## L    DETAILED CLC RESULTS

We visualize the improvement of CLC between all language pairs in Fig. 4 to Fig. 8. The left sub-figure in each panel reports the baseline CLC scores, while the right sub-figure shows the absolute change after applying DCO. Warmer colors indicate higher CLC scores, and the delta plots highlight systematic gains across most language pairs.

Notably, DCO consistently improves CLC, with substantial gains not only between typologically similar languages such as English-Spanish, but also between distant ones. For instance, Arabic–Chinese and Hindi-Japanese improve by 15% and 13%, respectively, while the Korean-French pair gains a remarkable 14% increase on Llama-3.1-8B.

These results indicate that DCO not only raises overall knowledge consistency but also narrows the gap between distant language families.

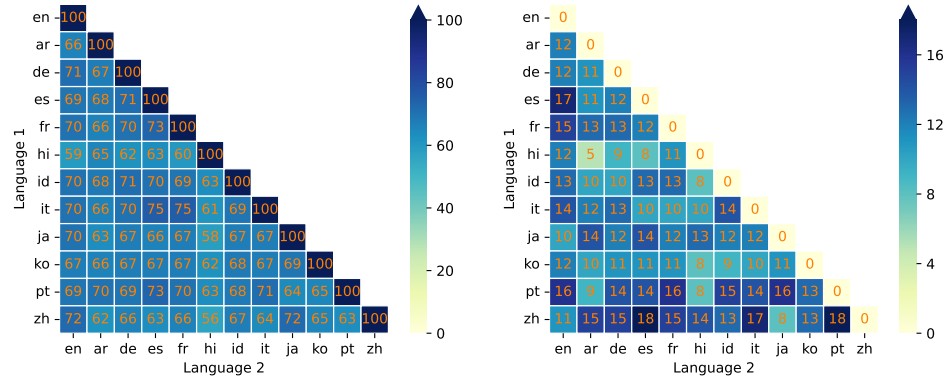

Figure 4: The changes in CLC of `Qwen2.5-14B` after DCO. Left: CLC between all language pairs on the original model. Right: Improvements in CLC of the post-DCO model.

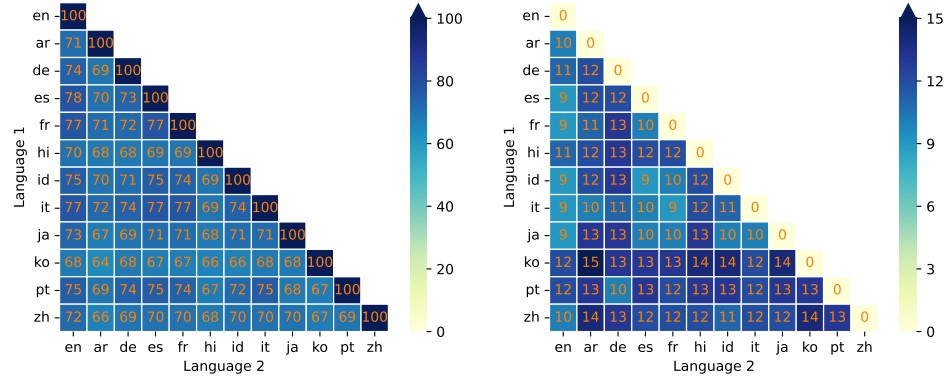

Figure 5: The changes in CLC of `Gemma3-12B` after DCO. Left: CLC between all language pairs on the original model. Right: The Improvements in CLC of the post-DCO model.

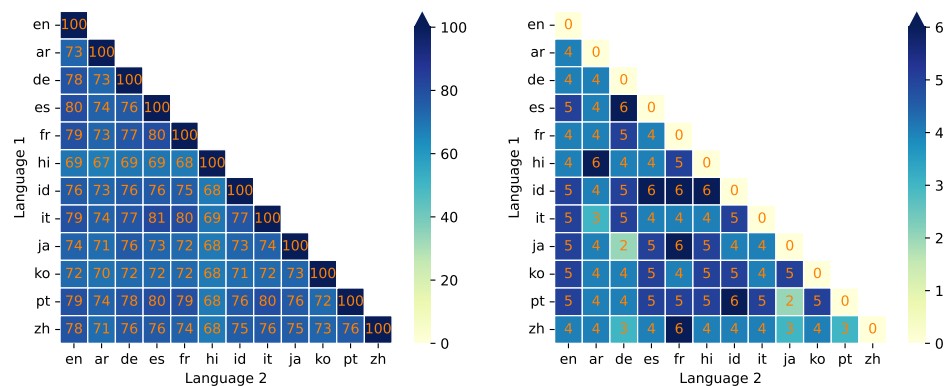

Figure 6: The changes in CLC of `Qwen3-14B` after DCO. Left: CLC between all language pairs on the original model. Right: The Improvements in CLC of the post-DCO model.

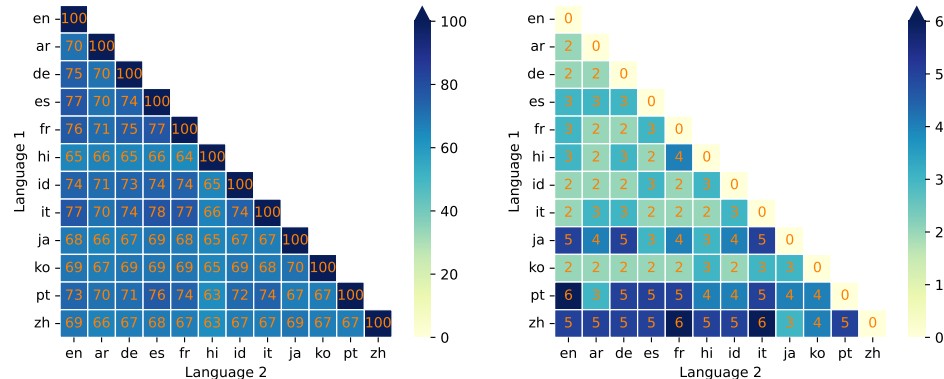

Figure 7: The changes in CLC of `Aya-Expanse-8B` after DCO. Left: CLC between all language pairs on the original model. Right: The Improvements in CLC of the post-DCO model.

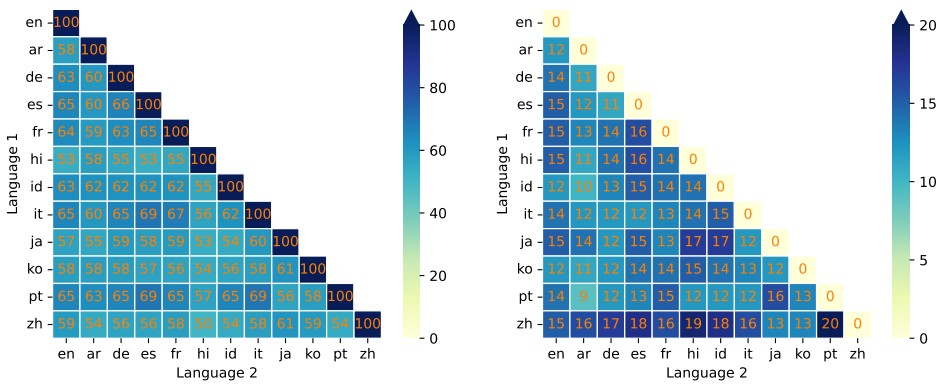

Figure 8: The changes in CLC of `Llama3.1-8B` after DCO. Left: CLC between all language pairs on the original model. Right: The Improvements in CLC of the post-DCO model.

## M    FULL BILINGUAL EXPERIMENTAL RESULTS

We present the detailed results of the bilingual experiments in Tab. 6 to Tab. 11

| | AR | DE | ES | FR | HI | ID | IT | JA | KO | PT | ZH |
|---|---|---|---|---|---|---|---|---|---|---|---|
| **Consistency** | | | | | | | | | | | |
| Qwen2.5-7B | 64.18 | 70.68 | 74.60 | 73.33 | 50.95 | 69.20 | 73.46 | 68.05 | 65.63 | 72.40 | 72.38 |
| + DCO | +9.17 | +9.20 | +6.68 | +8.88 | +14.66 | +7.42 | +7.50 | +8.16 | +8.01 | +9.99 | +7.66 |
| Qwen2.5-14B | 66.19 | 70.93 | 69.45 | 70.19 | 59.15 | 70.43 | 69.68 | 70.16 | 67.02 | 68.54 | 72.38 |
| +DCO | +12.09 | +12.82 | +14.68 | +14.11 | +12.98 | +11.04 | +15.66 | +9.15 | +10.80 | +15.76 | +9.54 |
| Gemma3-4B-pt | 63.21 | 68.24 | 71.78 | 71.32 | 56.84 | 64.48 | 70.53 | 58.28 | 60.73 | 69.00 | 60.75 |
| +DCO | +10.69 | +10.07 | +9.40 | +8.11 | +18.41 | +11.22 | +8.73 | +14.09 | +11.32 | +4.85 | +14.08 |
| Gemma3-12B-pt | 70.56 | 73.89 | 77.93 | 76.81 | 69.82 | 75.23 | 76.87 | 72.96 | 68.13 | 74.86 | 72.11 |
| +DCO | +7.37 | +8.44 | +4.64 | +4.58 | +9.07 | +8.41 | +6.11 | +6.79 | +9.91 | +7.93 | +5.45 |
| Qwen3-8B | 65.17 | 71.58 | 75.31 | 72.37 | 65.16 | 68.52 | 74.85 | 71.64 | 67.00 | 74.85 | 76.37 |
| +DCO | +9.48 | +9.77 | +7.17 | +9.10 | +4.80 | +10.61 | +7.43 | +5.80 | +7.68 | +7.73 | +2.99 |
| Qwen3-14B | 72.70 | 77.84 | 80.34 | 78.90 | 68.56 | 76.17 | 79.38 | 74.06 | 72.46 | 79.30 | 77.72 |
| +DCO | +3.73 | +4.83 | +4.50 | +4.86 | +4.08 | +5.29 | +4.66 | +6.03 | +4.89 | +5.10 | +4.75 |
| Aya-Expanse-8b | 69.66 | 74.58 | 77.42 | 76.26 | 65.44 | 73.74 | 76.70 | 68.45 | 68.69 | 73.49 | 69.39 |
| +DCO | +5.37 | +4.70 | +4.01 | +5.17 | +5.23 | +4.17 | +4.86 | +7.41 | +5.04 | +6.59 | +6.06 |
| Llama3.1-8B | 58.31 | 62.70 | 65.23 | 63.83 | 52.67 | 62.54 | 65.38 | 56.91 | 57.59 | 65.15 | 59.31 |
| +DCO | +10.75 | +10.94 | +12.57 | +13.89 | +14.88 | +7.99 | +6.21 | +14.50 | +11.07 | +13.73 | +16.14 |
| Llama3.2-3B | 46.07 | 54.38 | 54.06 | 53.65 | 42.78 | 51.77 | 53.25 | 45.11 | 42.75 | 56.23 | 47.82 |
| + DCO | +16.54 | +12.60 | +16.36 | +17.28 | +18.01 | +16.39 | +16.53 | +17.38 | +17.27 | +17.03 | +18.97 |

Table 6: Consistency improvements with English for each language across all models on MMMLU.

| | EN | AR | DE | ES | FR | HI | ID | IT | JA | KO | PT | ZH |
|---|---|---|---|---|---|---|---|---|---|---|---|---|
| **Accuracy** | | | | | | | | | | | | |
| Qwen2.5-7B | 69.57 | 54.51 | 59.98 | 63.43 | 62.30 | 44.33 | 58.71 | 62.66 | 59.07 | 57.13 | 61.93 | 63.98 |
| + DCO | -0.01 | +1.24 | +2.02 | +0.72 | +1.46 | +4.32 | +1.80 | +1.76 | +1.74 | +1.44 | +1.97 | +1.50 |
| Qwen2.5-14B | 72.46 | 54.88 | 59.52 | 56.83 | 58.31 | 45.66 | 61.06 | 56.21 | 63.24 | 56.96 | 56.35 | 69.87 |
| +DCO | +1.64 | +7.63 | +8.51 | +12.94 | +10.67 | +8.41 | +6.31 | +12.57 | +4.16 | +8.00 | +13.38 | +0.83 |
| Gemma3-4B-pt | 56.09 | 43.14 | 49.02 | 52.51 | 50.51 | 39.64 | 48.47 | 49.64 | 42.78 | 45.39 | 50.25 | 46.56 |
| +DCO | +0.08 | +1.04 | +2.77 | -0.02 | +0.26 | +5.32 | +2.93 | +1.60 | +4.64 | +2.13 | +1.81 | +3.07 |
| Gemma3-12B-pt | 70.07 | 58.60 | 64.19 | 65.43 | 64.14 | 57.94 | 62.07 | 65.05 | 60.83 | 60.49 | 65.01 | 61.36 |
| +DCO | -0.90 | +1.71 | +1.22 | -0.70 | +0.73 | +4.44 | +1.39 | +0.54 | +1.46 | +1.51 | +1.28 | +1.33 |
| Qwen3-8B | 71.15 | 56.34 | 63.41 | 65.60 | 64.58 | 54.29 | 60.99 | 64.78 | 61.32 | 59.47 | 64.11 | 66.29 |
| +DCO | +0.72 | +3.03 | +2.69 | +2.08 | +1.84 | +1.66 | +2.89 | +1.94 | +1.42 | +1.74 | +2.87 | +0.31 |
| Qwen3-14B | 76.58 | 63.07 | 68.56 | 71.04 | 70.10 | 58.69 | 67.49 | 70.35 | 65.65 | 64.74 | 70.72 | 69.64 |
| +DCO | +0.13 | +0.80 | +2.36 | +1.38 | +1.41 | +1.98 | +2.09 | +1.50 | +2.10 | +1.52 | +1.68 | +1.55 |
| Aya-Expanse-8B | 59.76 | 50.75 | 54.10 | 56.36 | 55.71 | 46.20 | 53.14 | 54.62 | 51.69 | 51.32 | 55.63 | 52.57 |
| +DCO | +0.52 | +0.46 | -0.01 | -0.28 | -0.03 | +0.96 | +0.57 | +0.66 | +1.37 | +0.01 | +0.72 | +0.68 |
| Llama3.1-8B | 57.27 | 41.19 | 49.13 | 51.17 | 47.56 | 35.01 | 46.88 | 49.89 | 44.63 | 43.10 | 48.71 | 46.40 |
| +DCO | +0.74 | +3.33 | +3.21 | +4.16 | +5.07 | +0.23 | +3.39 | +0.32 | +2.12 | +2.91 | +4.82 | +3.54 |
| Llama3.2-3B | 52.23 | 35.45 | 43.86 | 46.25 | 44.67 | 33.92 | 42.97 | 43.80 | 34.62 | 36.16 | 45.07 | 38.41 |
| + DCO | +0.93 | +4.64 | +2.48 | +1.75 | +2.91 | +3.17 | +1.66 | +2.83 | +5.96 | +4.42 | +2.16 | +4.90 |

Table 7: Accuracy of each model on MMMLU across all languages after DCO.

# N    FULL RESULTS OF OUT-OF-DOMAIN EXPERIMENTS

We present the full results on all five out-of-domain tasks in Tab. 12 and Tab. 13. The improvement in both evaluation dimensions suggests the significant generalizability of DCO.

| | ZH | DE | ES | FR | IT | JA | NL | PL | PT | RU | AR | VI | HI | SW | UR |
|---|---|---|---|---|---|---|---|---|---|---|---|---|---|---|---|
| **Consistency** | | | | | | | | | | | | | | | |
| Qwen2.5-7B | 64.41 | 72.26 | 77.82 | 71.57 | 71.96 | 57.78 | 65.49 | 65.41 | 75.27 | 66.67 | 65.49 | 68.80 | 49.76 | 35.58 | 48.02 |
| + DCO | +10.81 | +4.33 | +4.50 | +6.55 | +8.78 | +9.73 | +6.33 | +5.97 | +6.34 | +8.33 | +6.32 | +8.25 | | -0.28 | +7.83 |
| Qwen2.5-14B | 66.94 | 70.90 | 73.82 | 71.60 | 73.58 | 64.55 | 68.82 | 65.03 | 74.13 | 64.31 | 64.86 | 64.41 | 53.68 | 39.50 | 52.57 |
| + DCO | +8.03 | +5.80 | +8.21 | +5.93 | +3.81 | +5.92 | +8.52 | +8.34 | +6.52 | +8.73 | +11.00 | +10.68 | +5.33 | +0.40 | +4.90 |
| Gemma3-4B-pt | 54.23 | 60.91 | 57.85 | 56.83 | 60.28 | 50.21 | 56.93 | 53.29 | 57.68 | 57.18 | 53.50 | 52.78 | 51.56 | 37.36 | 47.26 |
| + DCO | +4.76 | +3.34 | +2.91 | +3.56 | +4.05 | +7.43 | +5.72 | +4.87 | +4.18 | +4.31 | +6.91 | +6.42 | +6.95 | +9.48 | +10.98 |
| Gemma3-12B-pt | 64.61 | 65.09 | 64.18 | 61.80 | 63.32 | 55.11 | 62.31 | 56.16 | 65.07 | 58.16 | 50.77 | 56.15 | 54.15 | 46.19 | 51.06 |
| + DCO | +3.40 | +3.27 | +6.15 | +4.54 | +4.88 | +3.51 | +4.99 | +5.94 | +1.54 | +5.76 | +7.04 | +5.72 | +3.35 | +0.65 | +8.45 |
| Qwen3-8B | 64.88 | 65.84 | 67.75 | 68.77 | 70.95 | 58.49 | 64.60 | 63.01 | 69.41 | 60.12 | 61.89 | 65.46 | 55.63 | 37.42 | 49.25 |
| + DCO | +4.42 | +6.58 | +8.88 | +5.64 | +4.60 | +4.19 | +7.06 | +6.84 | +6.63 | +7.25 | +10.64 | +5.01 | +4.52 | +5.19 | +5.69 |
| Qwen3-14B | 66.18 | 66.33 | 66.11 | 67.36 | 70.93 | 60.14 | 63.83 | 61.64 | 67.19 | 65.90 | 64.08 | 64.07 | 55.00 | 37.59 | 52.26 |
| + DCO | +4.36 | +8.36 | +12.06 | +7.23 | +3.29 | +6.19 | +9.93 | +8.55 | +8.70 | +4.54 | +5.40 | +7.32 | +6.53 | +4.88 | +9.70 |
| Aya-Expanse-8B | 66.36 | 69.33 | 72.63 | 71.18 | 69.02 | 57.02 | 67.24 | 64.38 | 65.18 | 64.96 | 65.29 | 65.77 | 59.01 | 35.93 | 45.34 |
| + DCO | +4.79 | +4.90 | +6.43 | +5.29 | +6.59 | +6.92 | +7.01 | +8.14 | +11.86 | +3.99 | +6.21 | +4.15 | +6.83 | +4.19 | +8.07 |
| Llama3.1-8B | 59.24 | 67.62 | 67.08 | 67.90 | 65.36 | 56.66 | 65.25 | 63.13 | 68.35 | 60.26 | 58.52 | 64.64 | 55.40 | 36.92 | 47.12 |
| + DCO | +11.29 | +3.89 | +12.32 | +5.15 | +8.13 | +6.75 | +5.51 | +5.56 | +4.56 | +9.53 | +6.21 | +2.45 | +14.60 | +24.20 | +16.28 |
| Llama3.2-3B | 57.52 | 64.49 | 65.19 | 62.84 | 61.03 | 53.57 | 59.29 | 54.31 | 61.62 | 60.54 | 59.98 | 57.48 | 52.58 | 35.28 | 45.53 |
| + DCO | +10.30 | +6.09 | +8.24 | +10.39 | +8.24 | +8.28 | +3.54 | +9.87 | +9.23 | +4.94 | -1.01 | +10.97 | +7.92 | +5.51 | +8.68 |

Table 8: Consistency improvements with English for each language across all models on XCSQA.

| | EN | ZH | DE | ES | FR | IT | JA | NL | PL | PT | RU | AR | VI | HI | SW | UR |
|---|---|---|---|---|---|---|---|---|---|---|---|---|---|---|---|---|
| **Accuracy** | | | | | | | | | | | | | | | | |
| Qwen2.5-7B | 84.00 | 55.50 | 58.00 | 66.00 | 60.50 | 62.00 | 52.50 | 54.50 | 57.50 | 63.00 | 55.50 | 53.50 | 61.50 | 39.50 | 25.00 | 41.00 |
| + DCO | -1.93 | +7.50 | +5.00 | +1.50 | +2.50 | +5.00 | +4.50 | +5.00 | +6.50 | +4.50 | +3.00 | +5.00 | +1.50 | +7.00 | 0.00 | +4.00 |
| Qwen2.5-14B | 87.00 | 59.50 | 62.00 | 65.00 | 62.50 | 65.00 | 57.50 | 61.00 | 60.00 | 67.50 | 56.00 | 54.50 | 56.50 | 46.50 | 32.50 | 47.00 |
| + DCO | -2.53 | +7.00 | +5.00 | +5.00 | +6.00 | +2.00 | +5.00 | +5.00 | +3.00 | +4.00 | +5.00 | +8.00 | +10.00 | +1.50 | 0.00 | +3.50 |
| Gemma3-4B-pt | 56.50 | 35.50 | 41.00 | 45.00 | 45.00 | 42.50 | 32.00 | 40.00 | 39.00 | 34.50 | 39.00 | 36.50 | 43.00 | 30.50 | 27.00 | 31.00 |
| + DCO | +2.30 | +5.50 | +3.50 | +0.50 | +0.50 | +4.00 | +3.50 | +4.00 | +1.00 | +3.50 | -0.50 | +2.50 | 0.00 | +2.50 | +1.50 | +3.50 |
| Gemma3-12B-pt | 66.00 | 52.50 | 52.50 | 53.50 | 52.00 | 56.50 | 43.00 | 53.00 | 45.50 | 53.50 | 43.00 | 39.50 | 46.50 | 40.00 | 37.50 | 40.00 |
| + DCO | +0.10 | +4.00 | +3.50 | +2.50 | +4.00 | +1.00 | +1.50 | +1.50 | +5.00 | +2.50 | +6.00 | +6.00 | +5.00 | +2.00 | +1.50 | +7.50 |
| Qwen3-8B | 73.50 | 53.00 | 53.50 | 56.50 | 61.00 | 60.50 | 49.00 | 55.50 | 53.50 | 57.00 | 51.50 | 50.50 | 56.50 | 47.00 | 28.00 | 41.50 |
| + DCO | +0.97 | +4.50 | +2.50 | +3.50 | -1.00 | +0.50 | -1.00 | +1.50 | +3.00 | +5.50 | +2.50 | +5.50 | +1.00 | +2.50 | +1.00 | +3.50 |
| Qwen3-14B | 77.50 | 55.00 | 58.00 | 60.00 | 57.00 | 62.50 | 51.50 | 59.00 | 57.00 | 59.00 | 57.00 | 55.50 | 60.50 | 47.50 | 27.50 | 43.00 |
| + DCO | +1.07 | +3.50 | +4.00 | +5.00 | +5.00 | +2.50 | +4.50 | +2.50 | +5.00 | +4.00 | +1.00 | +1.50 | +2.50 | +1.00 | +5.50 | +9.00 |
| Aya-Expanse-8B | 78.00 | 61.00 | 59.00 | 64.00 | 58.00 | 62.00 | 49.50 | 59.50 | 57.00 | 60.00 | 58.50 | 56.50 | 59.50 | 50.50 | 25.00 | 36.00 |
| + DCO | +0.57 | +4.00 | +2.50 | +5.00 | +4.50 | +3.00 | +5.50 | +1.00 | +4.00 | +7.00 | -2.00 | +2.50 | +4.00 | +3.50 | +3.50 | +7.00 |
| Llama3.1-8B | 67.50 | 48.00 | 55.50 | 54.50 | 55.00 | 56.50 | 41.00 | 52.00 | 51.00 | 53.00 | 48.50 | 47.00 | 55.50 | 39.50 | 27.00 | 32.00 |
| + DCO | +0.17 | +3.50 | 0.00 | +0.50 | +0.50 | -1.00 | +3.00 | -0.50 | +0.50 | +1.00 | +2.00 | -1.50 | +0.50 | +5.00 | +2.00 | +3.50 |
| Llama3.2-3B | 65.00 | 51.00 | 47.50 | 46.50 | 42.50 | 45.50 | 45.00 | 39.50 | 41.50 | 45.50 | 44.00 | 45.00 | 43.00 | 35.00 | 29.00 | 34.50 |
| + DCO | -4.07 | -3.50 | +3.00 | +8.50 | +6.00 | +3.50 | +4.00 | +2.50 | +8.50 | +3.50 | +2.00 | -1.00 | +8.00 | +4.50 | +1.50 | +4.00 |

Table 9: Accuracy of each model on XCSQA across all languages after DCO.

| | FR | NL | ES | RU | JA | ZH | KO | VI | EL | HU | HE | TR | CA | AR | UK | FA |
|---|---|---|---|---|---|---|---|---|---|---|---|---|---|---|---|---|
| **Consistency** | | | | | | | | | | | | | | | | |
| Qwen2.5-7B | 44.12 | 48.41 | 45.50 | 41.78 | 39.05 | 40.18 | 36.50 | 44.54 | 30.88 | 30.66 | 30.88 | 34.29 | 39.16 | 40.89 | 40.53 | 35.85 |
| + DCO | +17.90 | +13.98 | +16.21 | +15.08 | +16.89 | +11.53 | +18.40 | +19.05 | +17.79 | +15.54 | +17.27 | +17.10 | +14.64 | +13.24 | +15.02 | +14.15 |
| Qwen2.5-14B | 45.01 | 50.87 | 47.45 | 42.83 | 41.94 | 42.61 | 41.19 | 46.91 | 35.70 | 32.04 | 38.84 | 37.99 | 41.71 | 41.78 | 42.48 | 40.58 |
| + DCO | +17.33 | +14.48 | +14.35 | +16.05 | +16.66 | +9.77 | +16.23 | +16.32 | +16.48 | +14.32 | +15.93 | +13.36 | +15.40 | +15.40 | +16.67 | +14.38 |
| Gemma3-4B-pt | 38.26 | 48.78 | 45.63 | 40.57 | 36.99 | 31.78 | 35.72 | 45.23 | 38.16 | 34.85 | 35.91 | 39.77 | 36.88 | 35.51 | 43.75 | 35.92 |
| + DCO | +23.36 | +16.47 | +15.47 | +18.24 | +15.90 | +14.32 | +16.87 | +21.56 | +19.32 | +20.07 | +14.65 | +19.58 | +18.49 | +12.63 | +15.81 | +12.18 |
| Gemma3-12B-pt | 41.81 | 51.38 | 47.80 | 42.90 | 40.98 | 34.76 | 40.39 | 47.82 | 41.29 | 39.53 | 40.16 | 42.17 | 39.82 | 38.87 | 45.76 | 40.10 |
| + DCO | +23.27 | +15.53 | +15.53 | +17.37 | +15.17 | +15.12 | +15.74 | +20.81 | +19.02 | +18.83 | +13.72 | +19.46 | +17.92 | +11.74 | +15.76 | +11.35 |
| Qwen3-8B | 43.21 | 47.24 | 43.21 | 38.43 | 33.25 | 37.43 | 33.89 | 45.29 | 32.92 | 30.50 | 30.26 | 35.43 | 38.58 | 38.86 | 40.70 | 34.29 |
| + DCO | +15.21 | +14.19 | +13.47 | +14.05 | +16.92 | +11.13 | +16.15 | +13.71 | +14.91 | +16.06 | +12.46 | +14.27 | +13.44 | +11.47 | +14.45 | +13.91 |
| Qwen3-14B | 42.45 | 48.82 | 44.37 | 37.93 | 35.68 | 38.56 | 36.01 | 43.17 | 36.10 | 31.49 | 33.53 | 37.95 | 40.46 | 39.62 | 40.71 | 35.62 |
| + DCO | +16.43 | +14.11 | +15.01 | +17.66 | +20.46 | +12.22 | +16.78 | +17.53 | +15.39 | +18.40 | +14.80 | +14.37 | +15.12 | +13.71 | +18.13 | +18.09 |
| Aya-Expanse-8B | 50.81 | 51.38 | 58.03 | 39.84 | 39.72 | 38.32 | 36.77 | 55.08 | 34.74 | 29.12 | 36.97 | 41.80 | 37.02 | 40.11 | 44.69 | 36.43 |
| + DCO | +18.33 | +13.44 | +13.16 | +15.42 | +11.08 | +10.74 | +11.72 | +14.93 | +12.77 | +7.25 | +9.40 | +13.45 | +9.82 | +11.05 | +12.31 | +11.69 |
| Llama3.1-8B | 44.16 | 51.07 | 47.40 | 41.13 | 41.35 | 37.11 | 36.94 | 45.82 | 35.80 | 36.87 | 40.42 | 37.31 | 40.05 | 38.42 | 41.33 | 38.38 |
| + DCO | +18.83 | +13.48 | +14.66 | +15.91 | +12.74 | +16.33 | +15.32 | +19.62 | +19.85 | +16.26 | +10.56 | +17.47 | +19.42 | +11.64 | +17.89 | +11.23 |
| Llama3.2-3B | 43.00 | 47.98 | 43.92 | 38.72 | 37.89 | 32.10 | 34.73 | 44.14 | 34.21 | 32.74 | 38.04 | 33.17 | 39.14 | 37.21 | 38.65 | 35.67 |
| + DCO | +18.58 | +15.29 | +14.58 | +15.53 | +15.18 | +15.06 | +12.51 | +18.92 | +16.58 | +18.08 | +10.84 | +17.03 | +15.79 | +12.44 | +17.07 | +12.66 |

Table 10: Consistency improvements with English for each language across all models on BMLAMA.

| | EN | FR | NL | ES | RU | JA | ZH | KO | VI | EL | HU | HE | TR | CA | AR | UK | FA |
|---|---|---|---|---|---|---|---|---|---|---|---|---|---|---|---|---|---|
| | | | | | | | **Accuracy** | | | | | | | | | | |
| Qwen2.5-7B | 61.83 | 36.61 | 43.02 | 40.79 | 39.06 | 36.61 | 34.82 | 33.37 | 38.84 | 27.68 | 28.01 | 26.40 | 31.86 | 34.15 | 39.79 | 38.56 | 32.42 |
| + DCO | +5.61 | +19.31 | +14.57 | +15.68 | +12.61 | +16.57 | +14.23 | +17.75 | +15.35 | +12.61 | +12.56 | +13.89 | +12.73 | +13.23 | +9.32 | +13.34 | +10.88 |
| Qwen2.5-14B | 62.67 | 36.61 | 46.48 | 44.81 | 42.02 | 40.07 | 37.05 | 40.57 | 41.46 | 32.53 | 28.12 | 35.60 | 34.99 | 37.05 | 42.35 | 41.46 | 36.66 |
| + DCO | +6.33 | +21.20 | +12.90 | +12.56 | +13.67 | +15.18 | +14.28 | +15.85 | +13.12 | +15.85 | +13.34 | +12.89 | +14.62 | +11.17 | +15.12 | +13.51 | |
| Gemma3-4B-pt | 66.07 | 32.42 | 43.86 | 39.68 | 35.21 | 30.30 | 26.34 | 32.48 | 36.83 | 31.81 | 30.69 | 29.91 | 34.60 | 32.42 | 31.58 | 38.90 | 30.08 |
| + DCO | +2.16 | +22.55 | +14.73 | +19.58 | +19.31 | +18.81 | +17.63 | +17.19 | +19.14 | +18.02 | +19.59 | +15.74 | +17.97 | +19.76 | +15.13 | +16.57 | +16.01 |
| Gemma3-12B-pt | 68.25 | 37.56 | 49.00 | 43.14 | 38.28 | 36.77 | 30.75 | 38.11 | 39.90 | 38.11 | 35.88 | 37.00 | 37.05 | 36.22 | 36.44 | 41.91 | 36.38 |
| + DCO | +1.55 | +21.09 | +13.05 | +17.57 | +17.47 | +17.30 | +16.91 | +16.58 | +19.03 | +18.11 | +18.47 | +14.23 | +19.53 | +17.52 | +13.84 | +18.02 | +12.62 |
| Qwen3-8B | 58.37 | 36.83 | 42.08 | 38.06 | 37.17 | 33.15 | 32.25 | 32.70 | 38.50 | 30.36 | 26.17 | 26.23 | 32.37 | 33.82 | 37.22 | 42.69 | 29.52 |
| + DCO | +6.90 | +16.69 | +13.11 | +14.28 | +12.10 | +14.67 | +12.95 | +13.17 | +12.34 | +11.55 | +15.79 | +11.94 | +13.05 | +14.51 | +9.43 | +9.65 | +11.33 |
| Qwen3-14B | 58.43 | 34.99 | 44.87 | 40.23 | 38.45 | 36.72 | 33.20 | 33.43 | 37.05 | 34.88 | 28.35 | 31.81 | 34.71 | 37.05 | 39.17 | 42.63 | 34.49 |
| + DCO | +8.07 | +18.86 | +12.66 | +13.84 | +13.84 | +16.01 | +16.30 | +14.78 | +16.69 | +17.35 | +16.91 | +11.77 | +14.17 | +14.85 | +12.00 | +11.39 | +13.05 |
| Aya-Expanse-8B | 67.02 | 45.31 | 46.82 | 52.73 | 35.21 | 36.22 | 37.28 | 34.04 | 49.55 | 32.87 | 22.32 | 33.37 | 35.83 | 32.76 | 37.61 | 39.79 | 33.15 |
| + DCO | +1.43 | +14.68 | +11.55 | +9.44 | +18.31 | +12.27 | +11.16 | +13.90 | +12.11 | +8.54 | +10.16 | +14.67 | +10.32 | +10.55 | +12.05 | +13.78 | |
| Llama3.1-8B | 61.16 | 35.21 | 46.15 | 41.96 | 38.34 | 37.39 | 28.52 | 31.03 | 38.17 | 31.58 | 32.48 | 36.89 | 32.48 | 36.22 | 33.93 | 40.18 | 32.70 |
| + DCO | +7.17 | +23.61 | +14.73 | +17.58 | +17.80 | +14.34 | +22.43 | +18.19 | +21.26 | +19.87 | +17.58 | +9.82 | +18.41 | +20.64 | +12.50 | +18.36 | +14.73 |
| Llama3.2-3B | 61.05 | 35.38 | 43.08 | 38.67 | 34.60 | 32.31 | 24.67 | 26.95 | 35.55 | 29.35 | 28.74 | 35.16 | 27.57 | 35.21 | 32.25 | 36.22 | 28.24 |
| + DCO | +6.72 | +21.76 | +16.18 | +17.52 | +18.30 | +17.75 | +19.64 | +17.41 | +21.37 | +15.52 | +18.14 | +9.48 | +17.69 | +16.35 | +14.07 | +18.02 | +15.34 |

Table 11: Accuracy of each model on BMLAMA across all languages after DCO.

| Domain | AR | DE | ES | FR | HI | ID | IT | JA | KO | PT | SW | YO | ZH | BN |
|---|---|---|---|---|---|---|---|---|---|---|---|---|---|---|
| | | | | | | **Anatomy** | | | | | | | | |
| Base | 54.08 | 65.20 | 57.86 | 67.79 | 50.76 | 65.17 | 68.50 | 64.77 | 56.56 | 63.31 | 49.45 | 46.78 | 70.92 | 51.68 |
| + DCO | +13.10 | +7.02 | +17.56 | +12.04 | +9.30 | +13.69 | +11.42 | +11.12 | +10.49 | +19.53 | +4.81 | +3.20 | +12.00 | +7.91 |
| | | | | | | **Medical genetics** | | | | | | | | |
| Base | 66.91 | 78.76 | 75.42 | 71.46 | 56.92 | 74.52 | 74.54 | 77.25 | 66.76 | 77.88 | 62.25 | 59.95 | 79.97 | 62.66 |
| + DCO | +9.21 | +11.71 | +16.51 | +18.54 | +12.85 | +8.24 | +13.17 | +3.18 | +10.13 | +14.60 | +4.06 | +4.11 | +8.99 | +9.38 |
| | | | | | | **High school mathematics** | | | | | | | | |
| Base | 70.25 | 71.76 | 71.93 | 68.89 | 68.21 | 72.42 | 71.38 | 67.22 | 70.27 | 67.51 | 59.42 | 58.65 | 66.81 | 64.88 |
| + DCO | +8.59 | +8.55 | +14.72 | +13.07 | +10.86 | +13.30 | +11.21 | +12.76 | +9.48 | +16.02 | +8.12 | +4.15 | +14.26 | +12.67 |
| | | | | | | **College mathematics** | | | | | | | | |
| Base | 61.09 | 72.62 | 68.67 | 69.42 | 63.56 | 64.56 | 73.08 | 65.81 | 61.24 | 69.62 | 50.28 | 50.95 | 69.75 | 58.81 |
| + DCO | +13.72 | +7.71 | +17.22 | +13.31 | +8.55 | +14.01 | +10.30 | +8.80 | +10.38 | +13.61 | +12.08 | +7.26 | +13.39 | +8.68 |
| | | | | | | **High school world history** | | | | | | | | |
| Base | 83.50 | 83.06 | 87.21 | 86.67 | 74.73 | 81.53 | 87.19 | 83.20 | 54.06 | 85.53 | 56.25 | 44.98 | 41.79 | 74.35 |
| + DCO | +3.52 | +6.03 | +3.82 | +3.19 | +4.52 | +4.34 | +0.82 | +3.85 | +3.64 | +4.99 | -1.37 | +1.42 | +1.08 | +3.43 |

Table 12: Consistency with English under cross-domain settings using Qwen2.5-14B. The model is post-trained with data of 'high school microeconomics'.

| Domain | EN | AR | DE | ES | FR | HI | ID | IT | JA | KO | PT | SW | YO | ZH | BN |
|---|---|---|---|---|---|---|---|---|---|---|---|---|---|---|---|
| | | | | | | | **Anatomy** | | | | | | | | |
| Base | 68.89 | 43.70 | 46.67 | 45.93 | 53.33 | 34.07 | 50.37 | 51.85 | 57.04 | 44.44 | 43.70 | 32.59 | 31.85 | 72.59 | 40.00 |
| + DCO | +1.80 | 0.00 | +5.92 | +5.18 | +6.67 | +2.97 | +8.89 | +7.41 | 0.00 | +5.93 | +18.52 | -2.96 | -4.44 | -2.96 | +1.48 |
| | | | | | | | **Medical genetics** | | | | | | | | |
| Base | 82.00 | 61.00 | 73.00 | 66.00 | 66.00 | 46.00 | 74.00 | 63.00 | 79.00 | 59.00 | 69.00 | 53.00 | 53.00 | 78.00 | 53.00 |
| + DCO | +5.43 | +5.00 | +7.00 | +14.00 | +14.00 | +11.00 | +3.00 | +13.00 | -5.00 | +8.00 | +16.00 | +2.00 | -2.00 | +4.00 | +8.00 |
| | | | | | | | **High school mathematics** | | | | | | | | |
| Base | 53.70 | 43.70 | 45.56 | 39.63 | 45.19 | 43.33 | 42.96 | 39.26 | 48.52 | 47.04 | 43.70 | 34.81 | 30.74 | 60.00 | 40.74 |
| + DCO | +2.38 | +6.30 | +9.63 | +7.41 | +8.14 | +3.71 | +7.41 | +7.41 | +5.18 | +4.81 | +14.08 | +5.93 | +6.30 | -3.33 | +12.22 |
| | | | | | | | **College mathematics** | | | | | | | | |
| Base | 57.00 | 38.00 | 41.00 | 40.00 | 36.00 | 33.00 | 39.00 | 36.00 | 44.00 | 35.00 | 41.00 | 24.00 | 32.00 | 52.00 | 34.00 |
| + DCO | -0.36 | +11.00 | +15.00 | +8.00 | +22.00 | +10.00 | +11.00 | +18.00 | +16.00 | +11.00 | +13.00 | +15.00 | +4.00 | +10.00 | +9.00 |
| | | | | | | | **High school world history** | | | | | | | | |
| Base | 89.45 | 81.43 | 82.28 | 84.81 | 84.39 | 70.46 | 79.32 | 83.97 | 80.59 | 80.17 | 82.70 | 46.84 | 33.33 | 81.86 | 68.78 |
| + DCO | +0.55 | +1.27 | -0.85 | 0.00 | -0.85 | +0.43 | +1.69 | -1.69 | +1.27 | -2.95 | -0.42 | -2.96 | +0.85 | +0.42 | +2.53 |

Table 13: Accuracy under cross-domain settings using Qwen2.5-14B. The model is post-trained with data of 'high school microeconomics'.

