# OpenReview forum: "Optimizing Language Models for Crosslingual Knowledge Consistency"
_ICLR.cc/2026/Conference — Submitted to ICLR 2026_

### Official Review · Reviewer_JsVn · 2025-10-25

**Soundness:** 2
**Presentation:** 2
**Contribution:** 2
**Rating:** 4
**Confidence:** 4

**Summary:**

- The paper introduces DCO (Direct Consistency Optimizaiton), which is a RL inspired method as means for improving crosslingual knowledge consistency (CLC) in multlingual models. DCO differs from prior RLHF or DPO-based alignment methods by taking a structured reward function based o the log-likelihood agreement across translations of prompts and responses.
- DCO method proposed by the authors show that they theoretically guarantees consitency between cross-lingual knowledge when certain hyperparameter relations (as outlined in Lemma 1) hold.
- The authors evaluate DCO across 9 multilingual LLMs and 3 datasets and show that there are consistent improvements in both the CLC (RankC metric proposed in prior work) and non-English answer accuracy.
- They also conduct some ablation studies showing that DCO also performs well in bilingual settings (between English and one non-English language), showing promises in joint consistency improvements across English and other non-English languages (Section 6.2) and show out-of-domain transferability in Section 6.3 on different domains in the MMMLU dataset.

**Strengths:**

- In terms of originality, the authors propose a new formulation of CLC alignment as an RL problem with a piecewise reward function rather than using heuristic voting or representation intervention. DCO extends on the DPO-style training to multilingul consistency without human preference labels, which makes it a efficient method where there are no human preference datasets available. Also, DCO’s efficiency, in a sense that it doesn’t require explicit reward model or human annotation makes it practically adoptable for large-scale post-training.
- In terms of quality, the paper seems to have a clear theoretical derivation in terms of the reward definition, optimal policy and other proofs of consistency which made it easier to follow the concept of DCO.
- Figure 1 was especially helpful in understanding the intuition of how the authors tried to align the likelihoods across languages.

**Weaknesses:**

- The translation equivalence assumption made in Section 4.1 before Definition 1, where the authors assume the translation mappings of t1 and t2 seems to be strong, which may not directly hold for other open-ended or culturally grounded tasks beyond the short-form factual QA tasks used in the current paper. This questions the generalizability of the DCO method beyond short-form QA tasks where having consistent knowledge across languages might act as a stronger influence factor.
- Based on my understanding, it seems like DCO is operationalized almost entirely via RankC, which measures how similarly the model ranks candidate completions across languages. But then RankC depends on likelihood distributions over a fixed candidate set, where the candidate answers are parallelized across languages. So I believe RankC actually only evaluates agreement with oneself, not agreement with the truth. The paper tries to mitigate that by also reporting the accuracies of English and non-English languages, but those seem to be quite disjoint, since you can become “more consistent” while consistently wrong. This could be illustrated in scenarios where consistency improves even when English accuracy drops and non-English accuracy barely moves or even degrades in some settings (this pattern in observed for some cases in Table 1). This might be an evidence that “consistency” and “correctness” are not necessarily aligned.
- Further, RankC operates on multiple-choice or cloze-style completions with finite candidate sets. But in real-world deployment, many of the queries are open-ended and answers are free-form rather than users actually give a query and all multiple choice options. Thus, the metric seems to be rather measuring distributional alignment under very controlled conditions. Maybe one way to answer this is looking into qualitative bilingual judgments (are models actually saying the same thing in both languages, or just picking the same multiple-choice index?) or evaluate on open-ended factual QA tasks with no fixed options with human raters judging factual equivalence across languages.
- This is a pretty narrow baseline space relative to the current baselines available for multilingual alignment methods. To my knowledge, there is a large body of work on representation-space alignmen or causal interventions for multilingual consistency and transfer, which explicitly attempt to enforce that different languages activate similar internal features for the same factual content. The paper does mention these methods in the Related work section but does not empirically compare and without such comparisons, it seems hard to tell if DCO is actually better than these methods.
- The formal notion of consistency suggested by the authors in Equation 5 seems to rely more on the “ranking” equivalence but it does not account for any semantic equivalence or paraphrasing differences. This could bake in two limitations: one is that consistentcy is evaluated only over a fixed candidate set of completions that are assumed translationally aligned and second, consistency is more ordinal (which one do the LLM prefer more) than factual (is the LLM asserting the same proposition). The theory (Lemmas 1-2 and the product-of-experts optimal policy) is proved under this narrow notion of consistency, so the conclusions might not generalize for more realistic definition of consistency, such as factual equivalence in free-form answers.
- The paper lacks analysis in terms of training stability, sensitivity to hyperparameters or computational costs comparison with DPO or other baselines.
- All the tables in the main paper seems to be showing results in terms of English vs. aggregated results across all non-English languages. Since the number of languages of each tasks is quite large (14, 16, 17), having language-specific trends would provide a better understanding. I’m also curious whether this only works for a subset of the languages or if the gains still hold for lower-resource languages.
- The paper seems to only have pairwise comparisons between English and another non-English language, which also questions the generalizability to improving knowledge consistency between two non-English languages.

**Questions:**

- Are both XCSQA and BMLAMA entirely machine translated? The paper notes that MMMLU is translated by human annotators but no information was provided for the two other datasets. If so, does the machine translation quality impact the findings? What are the differences between machine translated dataset and human translated dataset? Would DCO also work for noisy translations?
- How expensive is DCO in practice compared to standard DPO in terms of compute and sample efficiency?
- Do the improvements hold consistently across typologically distant languages (e.g., Arabic to Chinese) versus closely related ones (e.g., Spanish to Italian)?
- The paper uses RankC for consistency, but have the authors considered or tested semantic metrics such as multilingual BERTScore or COMET? RankC relies on overlap over discrete candidates, which might not reflect deeper factual alignment.
- The paper claims DCO could generalize to “other consistencies” such as paraphrase or in terms of modality. Could you elaborate on what modifications would be needed to extend Equation 11 beyond the suggested bilingual lexical alignment?
- [Minor] The anonymous link in the abstract appears to end abruptly, but it’s unclear whether this was intentional for anonymization or an oversight.

---

> ### Author Response · Authors · 2025-11-21
> **Author Response [Part 1/4]**
>
> Dear Reviewer JsVn:
>
> We appreciate your detailed review, inspiring questions, and excellent suggestions!
>
> First, we would like to respond to the concerns raised in the Weakness section with some new on-policy RL experiments.
> > **[W1, W3, W5] Generalization to open-ended generation. Effectiveness on answer spaces beyond prescribed finite sets.**
>
> We conduct an on-policy RL experiment, which confirms **our method is effective on open-ended generation tasks**. The process can be briefly described as: (i) Prepare the open-ended generation questions in two languages; (ii) For each instance, the model is asked to generate a response; (iii) the response and the prompt are translated into the other language; (iv) the alignment reward $r_{align}$ can be computed based on the translated prompt and response; (v) model is optimized with the objective as shown in Eq. 7.
>
> The experiment is conducted on GSM8K, an open-ended math QA task, and MMMLU, where we allow reasoning before generating the final answer. We use Qwen2.5-7B-Instruct and Qwen2.5-14B-Instruct as the base policies, and Qwen3-4B as the translator, which can be replaced with any MT model in practice.
>
>
> **Training**: We train the models with the RL objective in Eq. (7), which contains a sequence-level reward as defined in Eq. (6). We do not train the critic model. A training batch consists of 32 questions. We generate two responses for each prompt, and use the reward difference as the advantage. The learning rate is set to 5e-6 for Qwen and 2e-6 for gemma, using the AdamW optimizer.
>
> **GSM8K (open-ended generation)**
> | Model | Acc EN | Acc ZH | Consistency |
> |---|---:|---:|---:|
> | Qwen2.5-7B-Instruct | 89.2 | 83.6 | 84.7 |
> | **+ on-policy RL** | **90.0** | **86.8** | **86.9** |
> | gemma-3-12b-it | 90.3 | 87.1 | 87.7 |
> | **+ on-policy RL** | **92.3** | **88.1** | **89.2** |
>
> **Analysis.** Even without any human-parallel pairs, on-policy DCO **improves both accuracy and consistency** for two different base models. This shows that DCO-style alignment can be applied to open-ended tasks by creating pseudo-pairs through translation.
>
> **MMMLU (allowing reasoning before final answer)**
> | Model | Acc EN | Acc ZH | Consistency |
> |---|---:|---:|---:|
> | Qwen2.5-7B-Instruct | 66.2 | 58.8 | 68.9 |
> | **+ on-policy RL** | **70.1** | **59.7** | **72.3** |
> | gemma-3-12b-it | 71.9 | 64.8 | 70.9 |
> | **+ on-policy RL** | **72.4** | **66.7** | **71.8** |
>
> **Analysis.** The same conclusion holds on MMMLU under open-ended RL training: DCO yields **consistent gains in both capability and CLC**, confirming that its applicability is not limited to settings with pre-well-defined cross-lingual pairs.
>
> ---
>
> > **[Q1] Are both XCSQA and BMLAMA entirely machine-translated?**
> > The paper notes that MMMLU is translated by human annotators but no information was provided for the two other datasets. If so, does the machine translation quality impact the findings? What are the differences between machine translated dataset and human translated dataset?
>
> **Translation sources.**
> X-CSQA is machine-translated from the English CSQA questions into multiple languages, and BMLAMA is derived from mLAMA and X-FACTR, a mix of machine-translated prompts (from mLAMA) with human-authored prompts (from X-FACTR). We will clarify this in the paper.
>
> **Impact of MT quality.**
> Our method shows consistent gains on both the human-translated benchmark (MMMLU) and the MT-based or hybrid ones (X-CSQA, BMLAMA). Also, our additional experimental results (reported below) using on-policy RL with Qwen3-4B as a translator also show consistent improvements on GSM8K and MMMLU. Thus, the effectiveness of our method is not dependent on high-quality human-translated data.
>
> **Human vs. MT benchmarks.**
> Human-translated data (MMMLU) is of higher quality and more natural to human readers. MT-based datasets (X-CSQA, BMLAMA) are noisier but offer broader multilingual coverage; we therefore treat them as complementary, and the fact that our gains hold across both types supports the robustness of our findings.
>
> ---
>
> > **[Q2] How expensive is DCO in practice compared to standard DPO in terms of compute and sample efficiency?**
>
> One part of the additional computational cost is the increased batch size due to the requirement of multilingual versions of a prompt and responses. However, this could be alleviated by training techniques such as gradient accumulation. In general, DCO does not add significant computational costs to DPO and can be done with the computational resources designed for training DPO.
>
> ---

---

> ### Author Response · Authors · 2025-11-21
> **Author Response [Part 2/4]**
>
> > **[Q3] Do the improvements hold consistently across typologically distant languages (e.g., Arabic to Chinese) versus closely related ones (e.g., Spanish to Italian)?**
>
> Thank you for the question. Yes, the improvements are **not limited to typologically close languages**. We ran DCO on **non-English pairs** in BMLAMA, covering both distant and less-distant pairs:
> (i) **Arabic–Chinese (ar–zh)** (typologically distant),
> (ii) **Korean–French (ko–fr)** (typologically distant),
> (iii) **English–Spanish (en–es)** (closer pair for reference).
>
>
> **BMLAMA Accuracy (%)**
> | Model | Acc en | Acc es | Acc ar | Acc zh | Acc ko | Acc fr |
> |---|---:|---:|---:|---:|---:|---:|
> | Base | 62.67 | 44.81 | 42.35 | 37.05 | 40.57 | 36.71 |
> | **DCO** | **68.97** | **57.37** | **50.33** | **48.49** | **52.46** | **53.68** |
>
> **BMLAMA Cross-lingual Consistency (CLC, %)**
> | Model | CLC (en–es) | CLC (ar–zh) | CLC (ko–fr) |
> |---|---:|---:|---:|
> | Base | 47.45 | 36.34 | 37.29 |
> | **DCO** | **61.80** | **46.74** | **52.24** |
>
>
> **Analysis.** DCO improves CLC and accuracy **across all tested language pairs**, including typologically distant ones. In particular, CLC increases by **+14.35% (en–es)**, **+10.40% (ar–zh)**, and **+14.95% (ko–fr)**, while **both languages’ accuracies rise in every pair**. Although distant pairs start from lower baseline consistency (expected due to translation/linguistic gaps), DCO still delivers substantial gains, especially between Korean and French, suggesting its effectiveness is **not dependent on typological similarity**.
>
> ---
>
> > **[Q4] The paper uses RankC for consistency, but have the authors considered or tested semantic metrics such as multilingual BERTScore or COMET? RankC relies on overlap over discrete candidates, which might not reflect deeper factual alignment.**
>
> Thank you for your great suggestion! If we understand correctly, the suggestion boils down to evaluating consistency using more sophisticated methods, thus to capture similarities beyond prescribed categories, which is indeed an interesting idea! We reported some new results on an on-policy RL experiment (see below), which allows reasoning before generating the final answer. These new results, to some extent, reflect the *deeper alignment* between the reasoning traces generated by the LMs, in the sense that semantically equivalent traces lead to the same answers.
>
> Our initial submission mostly focuses on factual alignment and its theoretical guarantees. As such, we choose a metric that well measures the effectiveness of improving cross-lingual answer consistency. Meanwhile, we also want to point out that the metric considers the distribution over the entire answer set. Unlike other existing methods, like Top1 alignment or correctness overlapping (introduced in Sec. 2), only consider the candidate with the highest likelihood.
>
> However, we agree with you that measuring deeper factual alignment will considerably strengthen the paper. We will investigate the feasibility of extending the consistency metrics to semantic ones in future work.
>
> ---

---

> ### Author Response · Authors · 2025-11-21
> **Author Response [Part 3/4]**
>
> > **[Q5] The paper claims DCO could generalize to “other consistencies” such as paraphrase or in terms of modality. Could you elaborate on what modifications would be needed to extend Equation 11 beyond the suggested bilingual lexical alignment?**
>
> Thank you for the question. We would like to emphasize that Eq. (11) is neither constrained to **bilingual** nor **lexical** alignment; it is a DPO-style reparameterization of Eq. (7), which is an objective of much broader alignment problems.
> In our current paper, Eq. (6) instantiates the reward using a bilingual translation operator $\tau$ and a reference LM likelihood, i.e., $r_{\text{align}}(x,y)=\gamma \log \pi_{\text{ref}}(\tau(y)\mid\tau(x))$.
> Eq. (11) then fits a reparameterized Bradley–Terry model $\hat r_\theta$ to match these cross-view log-likelihood ratios on paired samples.
>
> To extend the alignment to **other types of consistencies (e.g., paraphrase or modality)**, the **first** way is to adapt Eq. (11) to the task:
>
> 1. **Replace the translation operator with a generic “consistency transform”**
>    Let $g_j(\cdot)$ be any operator that maps a prompt/response into another *consistent view* $j$ (paraphrase, style transfer, modality conversion, etc.). Then define a generalized reward
>    $$
>    r_{\text{cons}}(x,y)=\sum_{j} \gamma_{ij} \log \pi_{\text{ref}}^{(j)} \big(g_j(y) \big| g_j(x)\big),
>    $$
>    where $ \pi_{\text{ref}}^{(j)} $ is a suitable reference scorer in view $j$ (often the same LM if the view is text). This is exactly the same algebraic role played by $\tau$ in Eq. (6); only the transform changes.
>
> 2. **Eq. (11) remains the same structurally; only the likelihood terms change.**
>    In Eq. (11), the log-ratios
>    $\log \frac{\pi_{\text{ref}}(y_w^2\mid x^2)}{\pi_{\text{ref}}(y_l^2\mid x^2)}$ and $\log \frac{\pi_{\text{ref}}(y_w^1\mid x^1)}{\pi_{\text{ref}}(y_l^1\mid x^1)}$ would be replaced by the corresponding ratios computed **after applying $g_j$**. The random pairing / “winner–loser free” construction of DCO does not depend on bilinguality.
>
> 3. **Handling non-bijective transforms (open-ended paraphrases or modality).**
>    If $g_j$ can produce multiple valid views (e.g., many paraphrases, multiple captions), we can sample $K$ views and use an expectation or log-sum-exp:
>    $$
>    \log \pi_{\text{ref}}^{(j)} (g_j(y)\mid g_j(x)) \approx \log\frac{1}{K}\sum_{k=1}^K
>    \pi_{\text{ref}}^{(j)} \big(g_{j,k}(y)\mid g_{j,k}(x)\big).
>    $$
>    This preserves the theoretical form of Eq. (6)/(11) while accommodating one-to-many alignment.
>
> Concretely:
> - **Paraphrase consistency:** set $g$ to a paraphraser (or back-translation pipeline) within the *same* language; $\pi_{\text{ref}}$ can remain the original LM. Then $r_{\text{ALIGN}}$ becomes a paraphrase-view likelihood reward, and Eq. (11) is unchanged except for swapping $\tau \rightarrow g$.
> - **Modality consistency:** set $g$ to a modality converter (e.g., ASR for speech→text, image captioner for image→text). The reference scorer $\pi_{\text{ref}}^{(j)}$ can be a text LM over the converted view, or a cross-modal model that provides a normalized log-score; Eq. (11) still applies with these log-scores in place of bilingual LM likelihoods.
>
> Besides, another way is to directly train the model with the objective function Eq. 6, where $r_{align}$ can be replaced with any reward model/functions.
>
> ---
>
> > **[Q6] [Minor] The anonymous link in the abstract appears to end abruptly, but it’s unclear whether this was intentional for anonymization or an oversight.**
>
> Thank you for your careful reading. We do it on purpose to ensure all links are anonymous in the paper. All models and code will be public for the non-anonymous version.
>
> ---

---

> ### Author Response · Authors · 2025-11-21
> **Author Response [Part 4/4]**
>
> > **[W2] ...RankC actually only evaluates agreement with oneself, not agreement with the truth. The paper tries to mitigate that by also reporting the accuracies of English and non-English languages, but those seem to be quite disjoint, since you can become “more consistent” while consistently wrong.**
>
> Thank you for noticing this important point. First, we agree with the reviewer that a model can be internally consistent yet still perform poorly, echoing the observation in [1] that **knowledge inconsistency is widespread even when accuracy is reasonable**.
>
> Second, we would respectfully note that prior work [1,2] has already established accuracy and consistency as two distinct, yet equally important, evaluation dimensions.
> Both properties are desirable for LMs, our goal is to improve consistency *without sacrificing accuracy*, i.e., to achieve a **Pareto improvement** where one dimension can be enhanced without harming the other. We demonstrate, both theoretically (Sec. 3) and empirically (Sec. 4), that such Pareto improvements are attainable even in the absence of gold labels. On the contrary, the examples in Table 1 pointed out by the reviewer, where consistency improves yet performance drops, happened to one of the baselines (CALM), illustrate a case of failure to obtain such an improvement.
>
> [1] Factual Consistency of Multilingual Pretrained Language Models, Fierro and Søgaard, 2022
> [2] Cross-Lingual Consistency of Factual Knowledge in Multilingual Language Models, Qi et al., 2023
>
> ---
>
> > **[W6] Training stability with different learning rates**
>
> Thank you for raising this point. We assess training stability by varying the learning rate (1e-5 vs. 1e-6). With lr=1e-5, DCO yields large and consistent improvements in both CLC and accuracy across all languages. With lr=1e-6, the model remains close to the base performance, i.e., gains are minimal, suggesting under-updating rather than instability or collapse. In our preliminary experiments, we tried learning rates in {1e-5, 5e-6, 2e-6}, which all led to improvements in accuracy and consistency. Overall, these results indicate that DCO is stable in training and works reliably under a reasonable learning rate range, while too low a rate may simply slow or limit the alignment progress.
>
>
> | Consistency | FR | NL | ES | RU | JA | ZH | KO | VI | EL | HU | HE | TR | CA | AR | UK | FA |
> |---|---:|---:|---:|---:|---:|---:|---:|---:|---:|---:|---:|---:|---:|---:|---:|---:|
> | Qwen2.5-14B | 45.01 | 50.87 | 47.45 | 42.83 | 41.94 | 42.61 | 41.19 | 46.91 | 35.70 | 32.04 | 38.84 | 37.99 | 41.71 | 41.78 | 42.48 | 40.58 |
> | DCO (lr=1e-5) | 62.34 | 65.35 | 61.80 | 58.88 | 58.60 | 52.38 | 57.42 | 63.23 | 54.56 | 48.52 | 53.16 | 53.92 | 55.07 | 57.18 | 59.15 | 54.88 |
> | DCO (lr=1e-6) | 45.99 | 51.98 | 48.06 | 43.46 | 42.79 | 43.13 | 41.64 | 47.57 | 36.73 | 32.95 | 39.90 | 38.72 | 42.47 | 42.49 | 43.15 | 41.38 |
>
> | Accuracy | EN | FR | NL | ES | RU | JA | ZH | KO | VI | EL | HU | HE | TR | CA | AR | UK | FA |
> |---|---:|---:|---:|---:|---:|---:|---:|---:|---:|---:|---:|---:|---:|---:|---:|---:|---:|
> | Qwen2.5-14B | 62.67 | 36.61 | 46.48 | 44.81 | 42.02 | 40.07 | 37.05 | 40.57 | 41.46 | 32.53 | 28.12 | 35.60 | 34.99 | 37.05 | 42.35 | 41.46 | 36.66 |
> | DCO (lr=1e-5) | 69.00 | 57.81 | 59.38 | 57.37 | 55.69 | 55.25 | 48.94 | 54.85 | 57.31 | 45.65 | 43.97 | 48.94 | 47.88 | 51.67 | 53.52 | 56.58 | 50.17 |
> | DCO (lr=1e-6) | 63.05 ​​| 38.17 | 47.54 | 45.09 | 42.75 | 40.68 | 37.00 | 40.90 | 42.63 | 33.31 | 28.91 | 36.83 | 35.49 | 37.83 | 42.63 | 42.75 | 37.61 |
>
>
>
> ---

---

> ### Author Response · Authors · 2025-11-27
> **Comparison with representation-based method**
>
> >[W4] Comparison with representation-based cross-lingual alignment
>
> We included new experiments on the KLAR dataset ([1]), comparing against a representation-based cross-lingual alignment baseline [1], which intervenes on hidden representations by taking a linear combination of representations of intermediate transformer layers. Our method improves on both accuracy and cross-lingual consistency compared to [1].
>
> | Model | Acc. (original) |   Acc. ([1]) | Acc. (DCO) |  CLC. ([1]) |   CLC. ([1]) | CLC. (DCO) |
> | --- |---|---|---|---|---|---|
> | bloom-560m | 43.24 | 51.67 |  **53.16** | 54.16 | 60.32  | **64.81** |
> | llama-2-7b | 71.47 | 76.08 | **80.90**  |  70.44 | 75.52  | **83.31** |
>
> The results of the baselines are taken from Tables 7 and 10 of [1], averaged over languages under study.
> Note that CLC is measured by the overlap ratio of correct predictions for parallel facts between language pairs following [1].
>
> [1] Lost in Multilinguality: Dissecting Cross-lingual Factual Inconsistency in Transformer Language Models, Wang et al., 2025.

---

### Official Review · Reviewer_qoeZ · 2025-11-02

**Soundness:** 4
**Presentation:** 4
**Contribution:** 3
**Rating:** 6
**Confidence:** 4

**Summary:**

The paper addresses the problem of crosslingual knowledge consistency (CLC) in large language models. The authors formulate this as a reinforcement learning problem and propose a novel, structured reward function, $r_{ALIGN}$, and introduces Direct Consistency Optimization (DCO), a practical, DPO-inspired algorithm that directly optimizes the policy to achieve this reward objective. Empirical results show that DCO significantly improves CLC, substantially outperforming the baseline.

**Strengths:**

DCO is theoretically grounded, and demonstrated excellent empirical performance in cross-language consistency. Notably, while optimizing consistency, DCO can also improve generalization without gold labels. Besides, DCO may extend to consistency optimization beyond cross-language consistency.

**Weaknesses:**

[W1] DCO relies on well-defined cross-lingual pairs. These pairs are readily available on datasets such as MMMLU, but not straightforward for general tasks such as dialogue and reasoning, which limits its applicability.

[W2] When gold labels are available, DCO improves cross-lingual consistency, but sometimes slightly degrades performance for English.

**Questions:**

[Q1] How does models trained using DCO and baseline methods perform when tested on datasets other than the training dataset (e.g., GSM8k)? This will characterize the degradation in LLM capability after cross-language consistency optimization.

---

> ### Author Response · Authors · 2025-11-21
> **Author Response [Part 1/2]**
>
> Dear Reviewer qoeZ:
>
> We appreciate your interesting and practical questions!
>
> ---
>
> > **[Q1] How do models trained using DCO and baseline methods perform when tested on datasets other than the training dataset (e.g., GSM8k)? This will characterize the degradation in LLM capability after cross-language consistency optimization.**
>
> Thank you for the question on cross-dataset cross-generalizability. To directly test whether DCO harms performance beyond multiple-choice training, we follow Reviewer qoeZ’s suggestion and evaluate on **GSM8K**, an open-ended math generation benchmark.
>
>
> Concretely, we train models on **MMMLU (en-zh or en-fr, multiple-choice)** and then evaluate **open-ended generation** on GSM8K in both English-Chinese and English-French setups (no GSM8K data is used during training).
>
>
> **Cross-Task Evaluation: GSM8K Accuracy (%) & CLC (%) for en-zh and en-fr**
> We use match rate as the consistency metric, *the percentage of questions to which the final numeric answers are the same across the two languages*.
> | Model | CLC(en-zh) | Acc en | Acc zh \| | CLC(en-fr) | Acc en | Acc fr |
> |--------------|-----------:|------------:|--------:|-----------:|------------:|--------:|
> | Qwen2.5-14B  | 69.37      | 84.84       | 74.30   \| | 62.70      | 84.84       | 65.88   |
> | SFT          | 69.14      | 82.79       | 75.36   \| | 63.00      | 82.79       | 67.55   |
> | DPO          | 66.64      | 86.28       | 71.11   \| | 66.03      | **86.28**       | **69.45**   |
> | CALM | 67.70      | 86.05       | 72.55    \| | 63.08  | 84.91       | 66.57   |
> | **DCO**      | **73.39**  | **87.11**   | **77.10**   \| | **66.87** | 86.05   | 69.29 |
>
> **Note: DPO and SFT used ground-truth data during training.*
>
> **Analysis.** These results indicate that DCO **does not degrade** cross-task capability; instead it yields **consistent gains in both accuracy and CLC** on open-ended generation. Relative to the base model, DCO improves EN–ZH CLC by **+4.02%** while also increasing accuracy on both English (**+2.27%**) and Chinese (**+2.80%**). The same pattern holds for EN–FR: DCO improves CLC by **+4.17%**, with accuracy gains on English (**+1.21%**) and French (**+3.41%**). In contrast, some baselines trade off consistency for accuracy (e.g., DPO improves EN accuracy but reduces EN–ZH CLC and ZH accuracy). **Overall, DCO provides the best “no-regret” cross-lingual alignment: higher consistency without sacrificing, and often improving, general task performance** on unseen open-ended datasets.
>
> ---

---

> ### Author Response · Authors · 2025-11-21
> **Author Response [Part 2/2]**
>
> >  **[W1] DCO relies on well-defined cross-lingual pairs, limiting applicability beyond datasets like MMMLU.**
>
> While explicit cross-lingual pairs are easiest to obtain in parallel QA datasets (e.g., MMMLU), **DCO is not restricted to multiple-choice** or to “given” bilingual pairs. Our general formulation in Sec. 4.3 defines a cross-lingual alignment reward that only requires *paired prompts/answers*, which can be **constructed on the fly** (e.g., in an on-policy RL procedure) for open-ended tasks via a lightweight translator.
>
> We conduct an on-policy RL experiment, which confirms **our method is effective on open-ended generation tasks**. The process can be briefly described as: (i) Prepare the open-ended generation questions in two languages; (ii) For each instance, the model is asked to generate a response; (iii) the response and the prompt are translated into the other language; (iv) the alignment reward r_{align} can be computed based on the translated prompt and response; (v) model is optimized with the objective as shown in Eq. 7.
>
> The experiment is conducted on GSM8K, an open-ended math QA task, and MMMLU, where we allow reasoning before generating the final answer. We use Qwen2.5-7B-Instruct and Qwen2.5-14B-Instruct as the base policies, and Qwen3-4B as the translator, which can be replaced with any MT model in practice.
>
> **Training**: We train the models with the RL objective in Eq. (7), which contains a sequence-level reward as defined in Eq. (6). We do not train the critic model. A training batch consists of 32 questions. We generate two responses for each prompt, and use the reward difference as the advantage. The learning rate is set to 5e-6 for Qwen and 2e-6 for gemma, using the AdamW optimizer.
>
>
> **GSM8K (open-ended generation)**
> | Model | Acc EN | Acc ZH | Consistency |
> |---|---:|---:|---:|
> | Qwen2.5-7B-Instruct | 89.2 | 83.6 | 84.7 |
> | **+ on-policy RL** | **90.0** | **86.8** | **86.9** |
> | gemma-3-12b-it | 90.3 | 87.1 | 87.7 |
> | **+on-policy RL** | **92.3** | **88.1** | **89.2** |
>
> **Analysis.** Even without any human-parallel pairs, on-policy DCO **improves both accuracy and consistency** for two different base models. This shows that DCO-style alignment can be applied to open-ended tasks by creating pseudo-pairs through translation.
>
> **MMMLU (allowing reasoning before final answer)**
> | Model | Acc EN | Acc ZH | Consistency |
> |---|---:|---:|---:|
> | Qwen2.5-7B-Instruct | 66.2 | 58.8 | 68.9 |
> | **+ on-policy RL** | **70.1** | **59.7** | **72.3** |
> | gemma-3-12b-it | 71.9 | 64.8 | 70.9 |
> | **+ on-policy RL** | **72.4** | **66.7** | **71.8** |
>
> **Analysis.** The same conclusion holds on MMMLU under open-ended RL training: DCO yields **consistent gains in both capability and CLC**, confirming that its applicability is not limited to settings with pre-defined cross-lingual pairs.
>
> **Remark.** Given the promising preliminary results on open-ended generation, we will dig into other types of consistency alignment like paraphrase (i.e., replacing the translator with paraphrase models for improving cross-paraphrase consistency). Nevertheless, we would like to note that this experiment is a preliminary attempt that demonstrates the method’s applicability; a more comprehensive study remains necessary in future work.
>
> ---
>
> > **[W2] When gold labels are available, DCO improves cross-lingual consistency, but sometimes slightly degrades performance for English.**
>
> Thank you for your careful reading. We also observe that the English accuracy is slightly degraded when adopting DCO on the Base model or at the top of DPO. We attribute this to the random data collection process of DCO, where the instances could be factually incorrect.
> However, the effect is **minor and not statistically significant (see updated PDF with standard error)**: overall, DCO **maintains or improves accuracy while boosting CLC** across models/datasets.
>
> Moreover, setting a lower **direction control ($\gamma$)* for English could mitigate the issue in practice (see Sec. 6.4), since it lets the model anchor the high-performance language (EN) while aligning others, an ability we analyze and motivate in the paper.
>
> ---

---

### Official Review · Reviewer_k1fx · 2025-11-09

**Soundness:** 2
**Presentation:** 3
**Contribution:** 3
**Rating:** 6
**Confidence:** 3

**Summary:**

This paper studies RL strategies for cross-lingual consistency gains. Specifically, the authors modified the first term of RLHF to align the ranking of the output answers (not exactly the distribution) for multiple languages. In experiments, the authors demonstrate that the presented method, DCO, can improve cross-lingual consistency in general settings, bilingual settings, and  OOD settings.

**Strengths:**

1. The authors provide sufficient evidence to support the claim.

2. The method is straightforward and well-motivated.
- The authors suggest aligning ranking instead of distributions, which is a good idea inspired by existing works.
- The method stems from RLHF and only makes minimal changes.

3. The authors show the method is compatible with other methods, e.g, DPO.

**Weaknesses:**

1.	Based on my understanding, the authors only conduct experiments on multiple-choice tasks. Can authors demonstrate some results on generation tasks, e.g., mLama? How do you configure BMLAMA? Do you treat it as a multiple-choice task or a generation task?

2.	I think authors should experiment with other tasks after DCO to show the method does not hurt performance on these tasks in an across-task generalization setting. Does DCO hurt the language modeling performance as it changes the ranking of output candidates?

**Questions:**

Please refer to Weaknesses

---

> ### Author Response · Authors · 2025-11-21
> **Author Response [Part 1/2]**
>
> Dear Reviewer k1fx,
>
> Thank you for your thoughtful feedback. We are glad that you found DCO well-motivated with solid empirical support, and that you appreciated our key idea of aligning cross-lingual **rankings** (rather than full distributions).
>
> We would like to first respond to **W2**, and then address **W1**.
>
> ---
>
> > **[W2] Does DCO hurt language modeling / cross-task performance by changing rankings?**
>
>
> Thank you for the question! To test whether DCO harms performance beyond multiple-choice training, we follow Reviewer qoeZ’s suggestion of doing a regression test on **GSM8K**, an open-ended generation math benchmark. Concretely, we train models on **MMMLU (en-zh or en-fr, multiple-choice)** and then evaluate **open-ended generation** on GSM8K in both en-zh and en-fr setups (no GSM8K data is used for training).
>
>
> **Cross-Task Evaluation: GSM8K Accuracy (%) & CLC (%) for en-zh and en-fr**
> | Model | CLC(en-zh)  | Acc en | Acc zh | CLC(en-fr)  | Acc en | Acc fr|
> |---|---:|---:|---:|---:|---:|---:|
> | Qwen2.5-14B | 69.37 | 84.84 | 74.30 | 62.70 | 84.84 | 65.88 |
> | DCO | 73.39 | 87.11 | 77.10 | 66.87 | 86.05 | 69.29 |
>
> **Consistency metric: match rate, the percentage of questions to which the final answers are the same across the two languages.*
>
> These results show that DCO **does not hurt** cross-task performance. Instead, it consistently **improves both accuracy and cross-lingual consistency** on open-ended generation across two bilingual settings. For en-zh, DCO yields +2.27% (en) and +2.80% (zh) accuracy gains over the base model, together with a +4.02% CLC improvement. For EN–FR, we observe the same pattern: DCO improves CLC by +4.17%, while also increasing accuracy on both English (+1.21%) and French (+3.41%). We attribute this to the effect DCO generalized to the mathematical domain of GSM8K.
>
> Overall, this indicates that the ranking-based knowledge alignment learned from MMMLU multiple-choice training generalizes reliably to GSM8K open-ended generation, benefiting consistency and task accuracy without using any GSM8K supervision.
>
> ---

---

> ### Author Response · Authors · 2025-11-21
> **Author Response [Part 2/2]**
>
> > **[W1] Based on my understanding, the authors only conduct experiments on multiple-choice tasks. Can authors demonstrate some results on generation tasks, e.g., mLama? How do you configure BMLAMA? Do you treat it as a multiple-choice task or a generation task?**
>
> Thank you for your question. Our method applies to general knowledge alignment, including open-ended generation. The simplified DCO is an efficient implementation of it when having a parallel dataset in hand. It only happens to be multi-choice tasks for most parallel QA datasets.
>
> We conduct an on-policy RL experiment, which confirms **our method is effective on open-ended generation tasks**. The process can be briefly described as: (i) Prepare the open-ended generation questions in two languages; (ii) For each instance, the model is asked to generate a response; (iii) the response and the prompt are translated into the other language; (iv) the alignment reward $r_{align}$ can be computed based on the translated prompt and response; (v) model is optimized with the objective as shown in Eq. 7.
>
> The experiment is conducted on GSM8K, an open-ended math QA task, and MMMLU, where we allow reasoning before generating the final answer. We use Qwen2.5-7B-Instruct and Qwen2.5-14B-Instruct as the base policies, and Qwen3-4B as the translator, which can be replaced with any MT model in practice.
>
> **Training**: We train the models using a sequence-level reward as defined in Eq. (6), and do not train the value model. A training batch consists of 32 questions. We generate two responses for each prompt, and use the reward difference as the advantage. The learning rate is set to 5e-6 for Qwen and 2e-6 for Gemma, using the AdamW optimizer.
>
> **Evaluation**: We use greedy decoding; the maximum number of tokens is set to 1024 for GSM8K and 128 for MMMLU. We measure consistency using answer match rate, the percentage of questions for which the final answers are the same across the two languages. The experiment shows that our method significantly improves accuracy and consistency:
>
> **GSM8K (open-ended generation)**
> | Model | Acc EN | Acc ZH | Consistency |
> |---|---:|---:|---:|
> | Qwen2.5-7B-Instruct | 89.2 | 83.6 | 84.7 |
> | **+ on-policy RL** | **90.0** | **86.8** | **86.9** |
> | gemma-3-12b-it | 90.3 | 87.1 | 87.7 |
> | **+ on-policy RL** | **92.3** | **88.1** | **89.2** |
>
>
> **MMMLU (allowing reasoning before final answer)**
> | Model | Acc EN | Acc ZH | Consistency |
> |---|---:|---:|---:|
> | Qwen2.5-7B-Instruct | 66.2 | 58.8 | 68.9 |
> | **+ on-policy RL** | **70.1** | **59.7** | **72.3** |
> | gemma-3-12b-it | 71.9 | 64.8 | 70.9 |
> | **+ on-policy RL** | **72.4** | **66.7** | **71.8** |
>
>
> **Remark**: Given the promising preliminary results on open-ended generation, we will dig into other types of consistency alignment like paraphrase (i.e., replacing the translator with paraphrase models for improving cross-paraphrase consistency). Nevertheless, we would like to note that this experiment is a preliminary attempt that demonstrates the method’s applicability; a more comprehensive study remains necessary in future work.

---

### Official Review · Reviewer_Rb3x · 2025-11-10

**Soundness:** 3
**Presentation:** 3
**Contribution:** 3
**Rating:** 6
**Confidence:** 4

**Summary:**

This paper introduces Direct Consistency Optimization (DCO), a DPO-inspired method for improving crosslingual knowledge consistency (CLC) in multilingual large language models (LLMs). Instead of relying on explicit reward models, DCO defines a structured reward function that encourages likelihood alignment between responses in different languages, achieving consistency without supervised labels. The paper presents theoretical guarantees for consistency, detailed empirical evaluations on 9 multilingual LLMs and 3 datasets (MMMLU, XCSQA, BMLAMA), and ablations exploring hyperparameter effects and generalization across domains.

**Strengths:**

- The problem of multilingual inconsistency is timely and practically relevant.
- The proposed reward formulation is mathematically grounded and connects neatly with DPO theory.
- The empirical evaluation is comprehensive, showing consistent improvements in CLC across many models and datasets.
- The bilingual and out-of-domain experiments strengthen the paper’s practical relevance.
- The direction-control parameters (γ₁, γ₂) provide an interpretable mechanism to balance alignment between high- and low-resource languages.

**Weaknesses:**

- The approach assumes accurate translation mappings and one-to-one correspondence between answers, which may not hold for open-ended or ambiguous tasks.
- Theoretical treatment for multi-language (>2) alignment is underexplored; practical guidance for setting γ parameters is limited.
- The experiments focus only on factual QA, leaving unclear how DCO performs in generative or paraphrastic tasks.
- There is little qualitative analysis of failure cases or discussion of translation noise sensitivity.
- Statistical significance and variance measures are missing from tables, limiting assessment of robustness.

**Questions:**

1. How robust is DCO when translation mappings are noisy or non-bijective?
2. Can the authors clarify how γ parameters should be tuned in practice for N > 2 languages?
3. Would likelihood normalization across languages improve stability?
4. Could DCO generalize to open-ended generation or paraphrase consistency tasks?

---

> ### Author Response · Authors · 2025-11-21
> **Author Response [Part 1/2]**
>
> Dear Reviewer Rb3x,
>
> Thank you for recognizing the practical importance of mitigating cross-lingual inconsistency and the theoretical novelty of our work! We answer your questions below:
>
> ---
> > **[Q1, Q4, W1, W3] Generalizability to open-ended consistency tasks, where crosslingual data can be non-bijective and the translation could be noisy.**
>
> Our framework applies to **open-ended generation**. And we agree that it’s important to adapt the knowledge alignment method to open-generation tasks, where a translator is used and the response data can be non-bijective.
>
> We conduct an on-policy RL experiment, which confirms **our method is effective on open-ended generation tasks**. We experiment on GSM8K, an open-ended math QA task, and MMMLU, where we allow reasoning before generating the final answer. We use two instruction-tuned models, Qwen2.5-7B-Instruct and gemma-3-12b-it, as the base policies, and Qwen3-4B as the translator.
>
> The process is: (i) Prepare the queries; (ii) The model generates a response; (iii) The response and the prompt are translated into the other language; (iv) compute alignment reward $r_{\text{align}}$ using the reference model.
>
> **Training**: We train the models using a sequence-level reward as defined in Eq. (6), and do not train the value model. A training batch consists of 32 questions. We generate two responses for each prompt, and use the reward difference as the advantage. The learning rate is set to 5e-6 for Qwen and 2e-6 for Gemma, using the AdamW optimizer.
>
> **Evaluation**: We use greedy decoding; the maximum number of tokens is set to 1024 for GSM8K and 128 for MMMLU. We measure consistency using answer match rate, the percentage of questions to which the final answers are the same across the two languages. The experiment shows that our method significantly improves accuracy and consistency:
>
>
> **GSM8K (open-ended generation)**
> | Model | Acc EN | Acc ZH | Consistency |
> |---|---:|---:|---:|
> | Qwen2.5-7B-Instruct | 89.2 | 83.6 | 84.7 |
> | **+ on-policy RL** | **90.0** | **86.8** | **86.9** |
> | gemma-3-12b-it | 90.3 | 87.1 | 87.7 |
> | **+ on-policy RL** | **92.3** | **88.1** | **89.2** |
>
> **MMMLU (allowing reasoning before final answer)**
> | Model | Acc EN | Acc ZH | Consistency |
> |---|---:|---:|---:|
> | Qwen2.5-7B-Instruct | 66.2 | 58.8 | 68.9 |
> | **+ on-policy RL** | **70.1** | **59.7** | **72.3** |
> | gemma-3-12b-it | 71.9 | 64.8 | 70.9 |
> | **+ on-policy RL** | **72.4** | **66.7** | **71.8** |
>
>
>
>
> **Remark**: Given the promising preliminary results on open-ended generation, we will dig into other types of consistency alignment like paraphrase (i.e., replacing the translator with paraphrase models for improving cross-paraphrase consistency). Nevertheless, we would like to note that this experiment is a preliminary attempt that demonstrates the method’s applicability; a more comprehensive study remains necessary in future work.
>
> ---
> > **[Q2, W2]. Can the authors clarify how γ parameters should be tuned in practice for N > 2 languages?**
>
>
> Thank you for the question! First, we would like to point out that the constraint on $\gamma$ when N > 2 is given in Appendix E, where $\gamma$ is generalized to an N by N matrix $\Gamma$, whose rank is 1.
>
> Also, we have discussed some principles of setting $\gamma$ in Appendix K. The same guidance can be naturally adopted for N>2 cases. Specifically, we emphasize that $\gamma$ controls the "*direction*" of consistency optimization, i.e., which language should the final LM be closer to: the easiest way is to **anchor on the highest-quality / highest-priority language** (EN in our study, but it could be any well-trained language depending on the application), and tune $\gamma$ to decide which languages remain stable vs. which shift toward the anchor.
>
> For **N > 2**, assuming $\beta=1$, this extends naturally by using a $\Gamma$ (or ratios w.r.t. an anchor):
> 1. **Choose an anchor language** $L_a$ (with highest quality / priority) and set $\gamma_{a,\ell}=1$.
> 2. **Initialize other languages** symmetrically (e.g., $\gamma_{\ell,a}=1$) when qualities are comparable or reciprocity is required; otherwise set **larger $\gamma$ for lower-quality / lower-resource languages** to encourage it move towards the anchor.
> 3. **Empirically tune on a small validation set**, using the **“ratio of changed answers” diagnostic** from Fig. 2, 3 and Appendix K:
>    - If a language is intended to be stable but it exhibits excessive changes, **decrease its $\gamma_{\ell,a}$**.
>    - If a language is expected to shift but it shows little movement, **increase its $\gamma_{\ell,a}$**.
>    This gives a simple feedback loop for multi-language tuning.
>
> We will add and highlight the analysis of this N-language $\gamma$-tuning procedure explicitly in the revised Appendix K.

---

> ### Author Response · Authors · 2025-11-21
> **Author Response [Part 2/2]**
>
> > **[Q3] Would likelihood normalization across languages improve stability?**
>
> We thank the reviewer for the suggestion. If we understand correctly, likelihood normalization refers to normalizing/scaling the magnitude of likelihoods across languages during training.
>
> For DCO, **shifting the absolute value** of per-language log-likelihoods does not affect the optimization: Eq. (11) is built on log-likelihood *differences* between preferred vs. dispreferred candidates and across languages, so changes in the absolute values cancel out in the objective.
>
> While **scaling the log-likelihoods by a coefficient** would correspond to a change in $\gamma$. This, in turn, is shown to be effective in Sec. 6.4, where we tested the alignment of English and two low-resource languages, Swahili and Yoruba.
>
> ---
>
> > **[W5] Statistical significance of reported results**
>
> Please see the updated PDF, which complies with ICLR submission guidelines. We have added standard deviations to the reported accuracy and consistency metrics in Tables 1 and 2. As shown in Table 1, except for the EN accuracies on aya-8b and Qwen3-14b, DCO significantly outperforms the unsupervised baselines and the base models across EN accuracy, non-EN accuracy, and consistency.
>
> ---

---

### Author Response · Authors · 2025-11-30
**General Response**

We thank all reviewers for their constructive feedback and for recognizing that our paper is
- **Timely and Relevant**: Reviewers agreed that mitigating cross-lingual inconsistency is a practically important and timely problem (Reviewers Rb3x).
- **Theoretically Grounded**: Our method, and the reward function we designed, are mathematically well-founded, connecting neatly with DPO theory (Reviewers Rb3x, k1fx, qoeZ, JsVn).
- **Effective**: The method is straightforward, requires no explicit reward model or human annotation, and demonstrates excellent empirical performance across diverse models and datasets (Reviewers k1fx, qoeZ, JsVn).
- **Interpretable**: The direction-control parameters $\gamma$ are interpretable for balancing alignment between languages (Reviewer Rb3x).

To address the reviewers’ concerns, during the rebuttal, we did

- **Open-Ended Generation**: We conducted new on-policy RL experiments on GSM8K (math generation) and MMMLU (with reasoning), which extend our method to open-generation and have shown its effectiveness there as well. We used **machine-generated** (Qwen3-4B) translations for validating the robustness of our method to translation noise, and the improvements still hold.
- **Cross-Task Generalization**: We performed regression tests using DCO-trained LMs on GSM8K and validated that our method does not hurt performance on tasks other than the trained tasks.
- **Comparison with Representation-Based Methods**: We added a comparison with a representation intervention baseline (Wang et al., 2025) on the KLAR dataset, showing our method achieves better accuracy and consistency.
- **Typological Generalization**: We provided additional results on BMLAMA for typologically distant language pairs (e.g., Arabic–Chinese, Korean–French), confirming DCO's effectiveness beyond closely related languages.
- **Statistical Significance & Stability**: We updated our results with standard deviations to demonstrate statistical significance and provided an analysis of training stability across different learning rates.

These new updates will be reflected in the final version of the paper. And all code and data will be made publicly available.

---

### Meta-Review · Area_Chair_vwB6 · 2025-12-31

**Summary:**

This paper proposes Direct Consistency Optimization (DCO), a reinforcement-learning–based approach for improving crosslingual knowledge consistency in multilingual large language models. By defining a structured reward that aligns likelihood rankings across languages, DCO aims to improve consistency without requiring explicit reward models or human annotations. The paper provides theoretical analysis connecting DCO to DPO-style optimization and reports empirical results on several multilingual QA benchmarks.

Across reviewers, the paper is generally regarded as well motivated, clearly presented, and technically sound, and the problem of crosslingual inconsistency is considered timely and practically relevant. The authors were responsive in the rebuttal, adding experiments on open-ended generation tasks, typologically distant language pairs, representation-based baselines, and reporting variance and stability analyses.

However, a key concern raised during review and discussion remains insufficiently resolved: the empirical advantage of DCO over strong baselines diminishes substantially on more recent and capable models. In particular, when evaluated on newer models such as Qwen-3, the reported improvements over existing methods are small and the comparison with one SOTA added during rebuttal was not conducted on newer models, raising questions about **the practical and scientific significance of the contribution for the current generation of LLMs**. As a result, while the method appears sound, the evidence does not convincingly support strong claims of effectiveness in the most relevant modern settings.

**Reviewer Concerns:**

Concerns that have been addressed satisfactorily:
- In response to concerns about limited evaluation beyond multiple-choice tasks raised by Reviewers Rb3x, k1fx, and JsVn: the authors conducted new on-policy reinforcement learning experiments on open-ended generation tasks (GSM8K and MMMLU with reasoning), demonstrating that the proposed method can be applied beyond fixed candidate settings and does not harm general task performance.
- In response to concerns about cross-task generalization and potential degradation of language modeling performance raised by Reviewers k1fx and qoeZ: the authors added regression experiments showing that models trained with DCO on MMMLU generalize to GSM8K without sacrificing accuracy, and in several cases improve both accuracy and crosslingual consistency.
- In response to concerns about typological generalization raised by Reviewer JsVn: the authors provided additional results on BMLAMA covering typologically distant language pairs (e.g., Arabic–Chinese, Korean–French), showing consistent gains across both distant and closely related languages.
- In response to concerns about training stability and variance raised by Reviewers Rb3x and JsVn: the authors added standard deviations, learning-rate sensitivity analyses, and stability discussions, indicating that DCO is stable under reasonable hyperparameter ranges.
- In response to concerns about translation noise and data sources raised by Reviewer JsVn: the authors clarified the use of human-translated versus machine-translated datasets and showed that the method remains effective under noisy translation settings, including when using Qwen-3 as a translator.

Concerns that have not been addressed satisfactorily:
- In response to concerns about comparison with representation-based baselines raised by Reviewer JsVn: the authors added experiments comparing against a representation-intervention baseline on the KLAR dataset, showing improvements in both accuracy and crosslingual consistency; **however, only two old base models are included; considering that the improvements on recent models as shown in Table 2 are consistently small, raising concerns about the practical significance of the proposed method for state-of-the-art LLMs.**

**Reviewer Scores:**

- Reviewer Rb3x: Marginally positive (6); indicated the paper could go either way.
- Reviewer k1fx: Marginally positive (6); raised concerns about practical impact.
- Reviewer qoeZ: Positive on soundness, but cautious about empirical significance.
- Reviewer JsVn: Below acceptance threshold (4); I think that Reviewr JsVn would not increase the score.

---

### Decision · Program_Chairs · 2026-01-26

Reject